# The added value of high resolution in estimating the surface mass balance in southern Greenland

Willem Jan van de Berg[1], Erik van Meijgaard[2], and Lambertus H. van Ulft[2]

[1]IMAU, Utrecht University, Utrecht, The Netherlands
[2]KNMI, De Bilt, The Netherlands

**Correspondence:** W. J. van de Berg
w.j.vandeberg@uu.nl

**Abstract.** The polar version of the regional climate model RACMO2, version 2.3p1, is used to study the effect of model resolution on the simulated climate and surface mass balance (SMB) of South Greenland for the current climate (2007-2014). The model data on resolutions of 60, 20, 6.6 and 2.2 km are intercompared and compared to SMB observations using three different data refinement methods: nearest neighbour, bilinear interpolation and a statistical downscaling method utilising the local dependency of fields on elevation. Furthermore, it is estimated how the errors induced by model resolution compare to errors induced by the model physics and initialisation.

The results affirm earlier studies that SMB components which are tightly linked to elevation, like runoff, can be refined successfully, as soon as the ablation zone is reasonably well resolved in the source dataset. Precipitation fields are also highly elevation dependent, but precipitation has no systematic correlation with elevation, which inhibits statistical downscaling to work well. If refined component-wise, 20 km resolution model simulations can reproduce the SMB ablation observations almost as well as the finer resolution model simulations. Nonetheless, statistical downscaling and regional climate modelling are complementary, the best results are obtained when high resolution RACMO2 data are statistically refined. Model estimates in the accumulation zone do not benefit from statistical downscaling; hence, a resolution of about 20 km is sufficient to resolve the majority of the accumulation zone of the Greenland Ice Sheet with respect to the limited measurements we have.

Furthermore, we demonstrate that using RACMO2, a hydrostatic model, at 2.2 km resolution lead to invalid results as topographic and synoptic vertical winds exceed 10 m/s, which violates the hydrostatic model assumptions.

Finally, additional tests show that model resolution is as important as properly resolving spatial albedo patterns, correctly initialising the firn column and uncertainties in the modelled precipitation and turbulent exchange.

## 1 Introduction

The Greenland Ice Sheet (GrIS) is the second largest ice sheet on Earth and in the most recent years the GrIS was the largest single contributor to global sea level rise (Vaughan et al., 2013; van den Broeke et al., 2016). The GrIS mass loss is partly due to enhanced glacial ice discharge, but most of the mass loss acceleration is caused by enhanced ablation, i.e. runoff of melt water from snow and ice melt. It is projected that in the future, enhanced ablation will increasingly negatively impact the mass balance of the GrIS (e.g. van Angelen et al. (2013); Fettweis et al. (2013)).

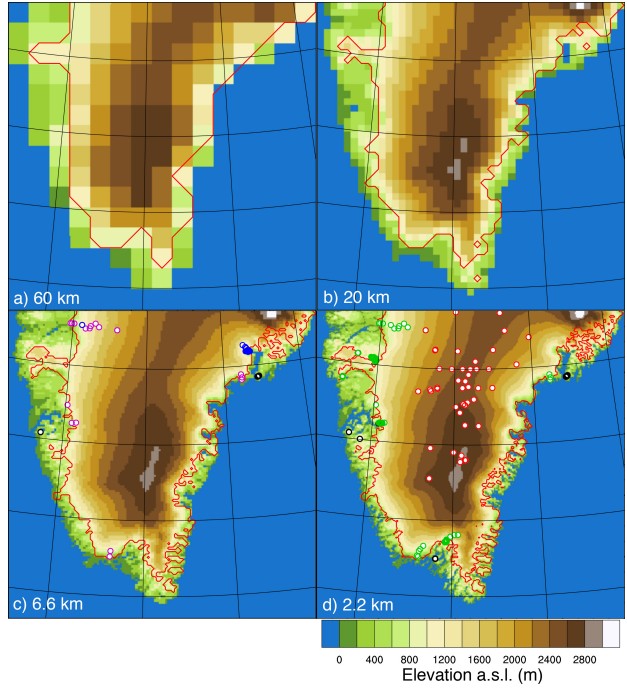

**Figure 1.** Model elevation and outline of glaciated areas (red) in the shared part of the domain for the 4 different model resolutions. In **(c)**, locations of the excluded (black), single year (magenta), summer (blue) time matched observations are drawn; In **(d)**, locations of excluded (black) and multi-year ablation (green) and multi-year accumulation (red) observations are drawn.

Runoff and precipitation are the two primary components of the (climatological) surface mass balance (SMB), which includes internal accumulation by melt water refreezing or retention. Undisputedly, the SMB depends strongly on the local topography. Precipitation fluxes generally decrease with elevation and latitude as the amount of atmospheric moisture is much smaller at lower temperatures. However, where topography blocks the atmospheric flow, precipitation is enhanced instead. Runoff increases with decreasing elevation, but it is not solely depending on elevation as, for example, precipitation provides a negative feedback on runoff through enlarged meltwater refreezing capacity and mitigated response of the melt-albedo feedback (e.g. Noël et al. (2015)). It would be a good presumption that resolving the topography is a prerequisite to resolve the spatial patterns of the SMB, but this has not been proven yet. However, the topography of Greenland complicates the evaluation of such presumption. Most of the interior ice sheet is rather flat and homogeneous (Fig. 1), because ice sheets level out elevation differences. Along the margins of Greenland, in contrast, the topography is rough and mountainous and the ice sheet and its adjacent glaciers are no longer fully covering the surface. Here, the horizontal length scales of the topography are small, 1 km or even less. In these marginal zones, the largest spatial gradients in SMB are expected.

Several methods exist to estimate the SMB of the GrIS (van den Broeke et al., 2017). Unfortunately, only in dry snow areas where the annual layering is preserved without internal accumulation, i.e. refreezing of percolating meltwater, can the SMB be

measured by remote sensing. For that reason, ice sheet encompassing SMB estimates can thus only be derived with numerical models. Earth System Models have made significant progress in including the complex physical processes that govern the atmosphere-glaciated surface exchange of mass and energy (e.g. Vizcaíno et al. (2013); van Kampenhout et al. (2019)). However, current conventional computational budgets limit ESMs to resolutions of $0.5°$ or $1.0°$, so South Greenland, for example, is resolved in comparable detail as shown in Figure 1a. To overcome this limitation it is common practice to use regional climate models (RCMs), i.e. dynamical downscaling, in analyzing the climate and SMB of the GrIS and typically resolutions at 20 to 5.5 km resolution are used (e.g. Noël et al. (2015, 2018); Fettweis et al. (2017); Langen et al. (2017)). The hydrostatic assumption used in these models inhibits a further refinement of the model grid, although some further refinement of the SMB can be archived by using fractional glacier masks (Fettweis et al., 2017). For simulations at higher resolutions non-hydrostatic (regional) models are required. This type of models, however, has been primarily developed for the purpose of weather forecasting in the mid-latitudes, so most of them yet lack the required detailed description of the relevant surface processes of glaciated surfaces. Nonetheless, non-hydrostatic models like WRF, NHM-SMAP and HARMONIE are increasingly applied for glaciated regions, e.g. Hines and Bromwich (2008); Mottram et al. (2017); Niwano et al. (2018); DuVivier and Cassano (2013), but due to their non-hydrostatic dynamical cores their computational costs are higher than hydrostatic models. Finally, SMB estimates up to the resolution of the latest digital elevation models (10 to 100 m) can be generated using statistical downscaling, as presented by Noël et al. (2016). For Greenland, a refinement to 1 km provided the best trade-off between data sizes and additionally resolved patterns. However, this type of statistical downscaling can only add value if the downscaled fields correlate locally with topography.

With three available methods, i.e. ESMs, dynamical downscaling and statistical downscaling, the question arises which (combination of) approach(es) provides the best estimate of local and integrated SMB of the GrIS and its peripheral glaciers. Are RCMs still required for the best possible SMB estimates or could statistical downscaling of ESM derived SMB provide equally good estimates? Furthermore, if RCMs still provide added value, what is the optimal resolution? For example, could a grid refinement of RCMs to kilometre scales provide superior results compared to all currently available methods? Finally, can we estimate for which resolution the approximations in model parameterisations and model initialisation become a source of error as important as model resolution?

In order to answer these questions, the polar version of the hydrostatic RCM RACMO2, version 2.3p1 has been run for South Greenland at four different resolutions: 60, 20, 6.6 and 2.2 km. These simulations, with and without statistical downscaling are analysed and compared to ablation and accumulation observations. The simulation at 2.2 km has been performed to investigate to which extent violating the assumptions made in RACMO2, i.e. hydrostatic atmospheric flow and that convective motion must be fully parameterized, deteriorate the model results. The results presented here, therefore, provide by no means any justification to apply a hydrostatic model on resolutions below 5 km on operational basis. Furthermore, the effect of grid refining on the SMB is compared to the effect of parameter tuning and modelling choices. In this way, the performance of statistical downscaling compared to dynamical downscaling can be assessed and the sources of uncertainty and errors can be determined, hence providing directions for further model development and research.

This manuscript starts with a description of RACMO2, some specific new parameterisations applied here and the observations used for evaluation. Next, the model results are analysed. First, statistical downscaling is applied to compare low-resolution RCM output with high resolution output, followed by a comparison of modelled data with observations. Then, we discuss the drivers of the differences in the model outcome for the highest resolution and finally, the sensitivity of model outcome to modelling and parameterisation choices is investigated. The manuscript is finalised with a discussion and conclusions.

## 2 Definitions, model, runs, statistical methods and observations

The climatological surface mass balance (SMB) is the local net mass gain or loss due to surface processes and internal accumulation, expressed in mm w.e. (water equivalents) $\mathrm{a}^{-1}$. For this study, we take the following components into account:

$$SMB = Prp - SU - RU - ER_{ds}, \tag{1}$$

where Prp, SU, RU and $ER_{ds}$ are precipitation, sublimation including drifting snow sublimation, melt water runoff and erosion due to transport of drifting snow, respectively. Precipitation is the primary mass source; SU, RU and $ER_{ds}$ are (predominantly) mass loss terms, for these three components positive values imply a mass loss.

### 2.1 RACMO2

The polar version of the Regional Atmospheric Climate Model RACMO2 has been used for over a decade to investigate the climate and SMB of Greenland. In the model simulations presented here, version 2.3p1 is used, which consists of the hydrostatic dynamics of the RCM HIRLAM, version 5.0.6 (Undén et al., 2002) and the physics package of the ECMWF IFS model, version cy33r1 (ECWMF-IFS, 2008). Additionally, the polar version incorporates a multilayer snow model (Ettema et al., 2010), including grain size dependent albedo (Kuipers Munneke et al., 2011), and snow drift (Lenaerts et al., 2012), to represent the specific atmosphere-surface interactions on ice sheets. Melt water percolation is modelled using the bucket method; if ice lenses are modelled in the firn pack, these are treated as permeable. Noël et al. (2015) presents an evaluation of modelled climate and SMB of the Greenland Ice Sheet of model version 2.3p1.

The parameterization of convection in the IFS physics uses an adapted version of the model presented by Tiedtke (1989). This module is used for all simulations, also for the run at 2.2 km resolution. The choice has drawbacks, as the parametization of convective clouds can start competing with the explicitly resolved mesoscale convective systems, reducing the quality of the model results. However, it should be kept in mind that this 2.2 km is run and discussed to show that RACMO2 is unsuitable for this resolution, irrespective of the quality of the modelled SMB. Furthermore, convective precipitation is of limited importance for the SMB the GrIS, where convection is generally weak and limited to summertime.

Turbulent exchange of heat and moisture of the surface with the lowest model layer, which is at approximately 10 m, are parameterised using Monin-Obukhov similarity theory (ECWMF-IFS, 2008). It assumes that, for neutral boundary layer conditions, the profiles of wind, dry static energy (heat) and specific humidity change logarithmically from their surface values

**Table 1.** List of variables in Equations (3) and (4).

| Symbol | Value | Unit | Description |
|--------|-------|------|-------------|
| $\rho$ | | kg m$^{-3}$ | Snow density |
| $\rho_i$ | 900 | kg m$^{-3}$ | Density of ice |
| $\rho_s$ | 600 | kg m$^{-3}$ | Threshold density for fast snow compaction |
| $E_k$ | $1.7 \cdot 10^4$ | J mol$^{-1}$ | Activation energy |
| $E_{c,\,\text{A10}}$ | $4.24 \cdot 10^4$ | J mol$^{-1}$ | Activation energy |
| $k_{c,\,\text{A10}}$ | $\begin{cases} 9.2 \cdot 10^{-9} & \rho \leq 550 \text{ kg m}^{-3} \\ 3.7 \cdot 10^{-9} & \rho > 550 \text{ kg m}^{-3} \end{cases}$ | kg$^{-1}$ m$^3$ s$^{-1}$ | Compression constant |
| $k_c$ | $2.45 \cdot 10^{-2}$ | Pa$^{-1}$ | Compression constant |
| $k_s$ | $9.3 \cdot 10^{-4}$ | m$^3$ kg$^{-1}$ Pa$^{-1}$ | Compaction constant |
| $r$ | | m | Effective grain size |
| $R$ | 8.314 | J mol$^{-1}$ K$^{-1}$ | Gas constant |
| $\sigma$ | | Pa | Overburden pressure |
| $t$ | | s | time |
| $T$ | | K | Snow temperature |

to their values at the lowermost model level. The boundary layer is generally stable over glaciated surfaces, in that case the logarithmic profiles are corrected with stability profile functions of similar shape as proposed by Holtslag and de Bruin (1988). In case of unstable conditions, the flux profiles of Dyer and Hicks (1970) are used. The transition height from near-surface laminar flow to turbulent flow and the effectiveness of turbulent exchange depends, among other factors, on the surface roughness length. For glaciated surfaces, the roughness length for momentum ($z_{0m}$) is set to 1 mm and 5 mm for snow covered and bare ice surfaces, respectively. The roughness length of heat ($z_{0h}$) and moisture ($z_{0q}$), a constant value for all other land surface types, is defined by the parameterisations of Andreas (1987) and Smeets and van den Broeke (2008b) for snow covered and bare ice surfaces, respectively. In these two parameterisations, ($z_{0h}$) and ($z_{0q}$) decrease for increasing turbulence, hence limiting the increase of turbulent exchange. These parameterisations represent the flux limiting effect of stratification on turbulent mixing while the surface drag (form drag) is less affected by stratification as it is also generated by the pressure fluctuations in the turbulent wake behind roughness elements (Smeets and van den Broeke, 2008b).

The reference polar version of RACMO2 employs the snow densification formulas as presented by Ligtenberg et al. (2011). These formulas, however, require precise *a priori* estimates of the local annual snowfall and temperature, which were not available prior to the simulations. Therefore, we initially explored an expression for creep of consolidated ice with cylindrical pores (Arthern et al., 2010, Eq. (B1)),

$$\frac{\partial \rho}{\partial t} = k_{c,\,\text{A10}}(\rho_i - \rho)\exp\left[-\frac{E_{c,\,\text{A10}}}{RT}\right]\sigma\frac{1}{r^2}. \tag{2}$$

The variables used in this and following Eqs. are listed in Table 1. However, we were unable to tune this relation to match the modelled firn densities with snow density profiles from Antarctica (van den Broeke, 2008). We chose to focus on Antarctic firn cores for tuning as in the Antarctic interior melt, which significantly alters the properties of firn cores and the densification process, is not occurring. Specifically, this equation fails to represent both the fast densification of low density, fine grained snow under very weak overburden pressure and the slower densification once the snow is denser and coarser grained while

the overburden pressure is orders of magnitudes bigger. However, densification partly depends on the recrystallisation of snow, which leads to a net growth of the crystals. Therefore, we modified Eq. (2) to the following empirical formula

$$\frac{\partial \rho}{\partial t} = k(\rho) \exp\left[-\frac{E_k}{RT}\right] \sigma \frac{1}{r^3} \frac{\partial r^3}{\partial t}, \tag{3}$$

in which $k(\rho)$ is defined as

$$
\begin{aligned}
\rho < \rho_s: \qquad & k(\rho) = k_c(\rho_i - \rho) + k_s(\rho_s - \rho)^2 \\
\rho \geq \rho_s: \qquad & k(\rho) = k_c(\rho_i - \rho).
\end{aligned}
\tag{4}
$$

In Eq. (3), $k(\rho)\exp\left[-E_k/RT\right]$ represents the strength of the snow structure; $\sigma$ is the effective pressure applied on the snow and $(1/r^3)(\partial r^3/\partial t)$ provides a scaled measure of recrystallisation rate. The modelled evolution of the snow grain size $r$ includes dry and wet snow metamorphism and is discussed in Kuipers Munneke et al. (2011). Equations (3) and (4) were tuned using snow density profiles from Antarctica (van den Broeke, 2008) and are able to reproduce these density profiles almost as good as the model presented by Ligtenberg et al. (2011).

## 2.2 Model simulations

We performed simulations on four domains with resolutions of approximately 60, 20, 6.6 and 2.2 km, respectively. These four domains have a shared interior on which no boundary relaxation conditions were applied. This interior, on which we analyse the results, covers 900×900 km, thus 15×15 and 405×405 grid boxes for the coarsest and finest model resolutions, respectively. The shared interior and the topography as resolved by the models are shown in Figure 1. Around this interior, all

140 domains include subsequently several rows of non-relaxed grid points surrounded by the boundary relaxation zone. Runs at the four resolutions were carried out with a time step of 150, 150, 150/90 and 60 s, respectively. For the 6.6 km simulation, the smaller time step of 90 s was only used for months with high wind speeds which causes the Lagrangian advection scheme to fail at processor sub-domain boundaries. In the 2.2 km simulation, this problem was mitigated by extending the shared-data rim around sub-domains from 6 to 8 grid boxes. Simulations at all resolutions were carried out on the same vertical mesh with

145 40 model levels. The lowest model levels were at approximately 10, 30 and 90 m above the surface.

Simulations cover the period September 2006 to October 2014 and ECMWF Operational Analyses were used as boundary conditions. These boundaries and simulation period were chosen because Operational Analyses have a spatial resolution of 25 km (February 2006 to January 2010) and 16 km (February 2010 onwards) for this period, which allows to drive the 2.2 km RACMO2 run without intermediate RACMO2 simulation. Although this 11-fold grid refinement step is bigger than typically

used for high-resolution studies, it was preferred here to a double-nesting approach, as the latter would inhibit comparing

simulations covering similar domains. The runs have not been extended to more recent years as the last observations used here for evaluation (discussed below) were conducted in the summer of 2014.

Since no *a priori* outline is known of the modelled ablation, percolation and dry snow zone, we chose to use a uniform initialisation of the snow model. All runs were initialised with a fresh snow layer of 50 cm in order to have, without long spin up, good results in the ablation zone where perennial snow is absent. For the dry snow zone this initialisation is also deemed sufficient because it reproduces the thermal characteristics of a thick snow layer. However, in the percolation zone this relatively thin snow pack lacks refreezing capacity, especially if precipitation is low. This affects runoff estimates in the percolation zone as will be discussed later. Finally, a constant ice albedo of 0.42 – a typical bare ice albedo value for the GrIS – is used, instead of MODIS derived albedo as is used in Noël et al. (2015), in order to improve the comparability of the model results on various resolutions. The reference soot concentration is set to 0.10 ppmv, equivalent to the default value in RACMO2.1 and RACMO2.3p1 (van Angelen et al., 2012; Noël et al., 2015).

Unless stated differently, the first year of the simulation is excluded to reduce the effect of firn layer spin-up on the modelled SMB.

## 2.3 Refinement methods

Besides using RACMO2, which can be seen as using dynamical downscaling as refinement method, three methods to refine output were tested: nearest neighbour remapping, bilinear interpolation and a form of statistical downscaling. We tested both downscaling of the modelled SMB as well as a component-wise downscaling of the SMB components in Eq. (1). All methods are applied on period-averaged accumulated quantities and not on daily accumulated SMB fields.

The SMB is only derived for ice sheet model grid points because outside this mask no valid estimates of SU, RU and $ER_{ds}$ are provided. For nearest neighbour estimates, we assume that SU, RU and $ER_{ds}$ are zero outside the ice sheet. For bilinearly interpolated estimates, modelled fields of SU, RU and $ER_{ds}$ are extrapolated outside the ice sheet domain prior to interpolation. The extrapolation was done iteratively by assigning the average of eight surrounding grid boxes to unassigned grid boxes if at least three of these grid boxes have an assigned value.

Statistical downscaling from a host domain to a target domain uses the local dependency on topography (Noël et al., 2016). Unlike Noël et al. (2016), no subsequent adjustment step, in which local differences in the bare ice albedo of the coarse and high-resolution grid are used to minimize the mean SMB bias, has been carried out. Here, the downscaling is set up in the following manner: On every point on the host domain the local regression of a property ($y$) to elevation ($h$) is derived using the 8 surrounding grid points, thus $y(h) = y_0 + hb$. This regression is only calculated if 4 or more of these 9 grid points have valid data. For Prp, all grid points are valid; for SMB, SU and $ER_{ds}$ only data points on the ice sheet are valid, and for RU, only ice sheet points with non-zero runoff are valid. Ice sheet points with zero RU are excluded, because otherwise the regression of RU to elevation would be underestimated. If the host point of interest has valid data, $y_0$ is chosen in such a way that $y(h)$ matches the modelled $y$ at the elevation of the grid point, otherwise $y_0$ is derived from the best fit. For RU, only negative values for $b$ are accepted, i.e. runoff decreases with altitude, a positive $b$ is set to zero. Next, $y_0$ and $b$ are extrapolated to host grid points

for which no regression could be derived. Finally, $y_0$ and $b$ are bilinearly interpolated to the target grid, and refined estimates
are derived using the local target elevation.

## 2.4 Observations

The modelled SMBs are compared with the ablation observations collected by Machguth et al. (2016b) and the accumulation
observations presented by Bales et al. (2001, 2009). From the various data sets, only observations on the common model
domain with known elevations were used.

The observations provided by Machguth et al. (2016b) are used in three ways, with within brackets the abbreviated name:

1. *(Sub)-annual and time matched [Annual ablation]*: Most of the ablation observations are precisely dated, so the ablation
   observations covering parts of the period between September 2006 and October 2014 are compared with model esti-
   mates for matching time periods. 128 observations at 40 sites fulfilled these criteria; 108 of them were annual ablation
   observations, one was accumulated ablation of 2 years and 19 were summer observations. These locations are shown in
   Figure 1c. The observations left out are on Mittivakket near Tasiilaq and Qasigiannguit glacier near Nuuk, both because
   the observations are too far away from glaciated points in the 6.6 and 2.2 km simulations.

2. *Averaged and time matched [Period ablation]*: In order to reduce the possible impact of misrepresented temporal vari-
   ability, the 128 observations at those 40 sites were averaged into 40 annual SMB estimates. The model estimates are
   again derived using the time period covered by these averaged annual SMB estimates.

3. *Averaged [Average ablation]*: Finally, as the short period for which RACMO2 is run excludes many observations, aver-
   aged modelled SMB for the October 2007 to September 2014 are compared to all available ablation observations, hence
   neglecting the interannual and decadal variability of SMB. For this purpose, separate records of ablation observations for
   a site were averaged into a single average per station, leading to 106 ablation estimates. Sites with only winter or sum-
   mer balances were excluded. The locations of the multi-year mean ablation and accumulation observations are shown
   in Figure 1d. Observations within the model domain but well outside the ice sheet masks at one of the resolutions were
   also excluded. These observations are shown in Figure 1d as well.

The accumulation observations, of which a considerable number was carried out well before 2006, are only used in the
evaluation of the averaged modelled SMB for the full 7-year period from October 2007 to September 2014. Furthermore,
accumulation observations with an elevation deviating more than 25 m from the Greenland Ice Mapping Project (GIMP, Howat
et al. (2014)) elevation were excluded as such a big elevation deviation for sites at the flat interior ice sheet indicate a location
error.

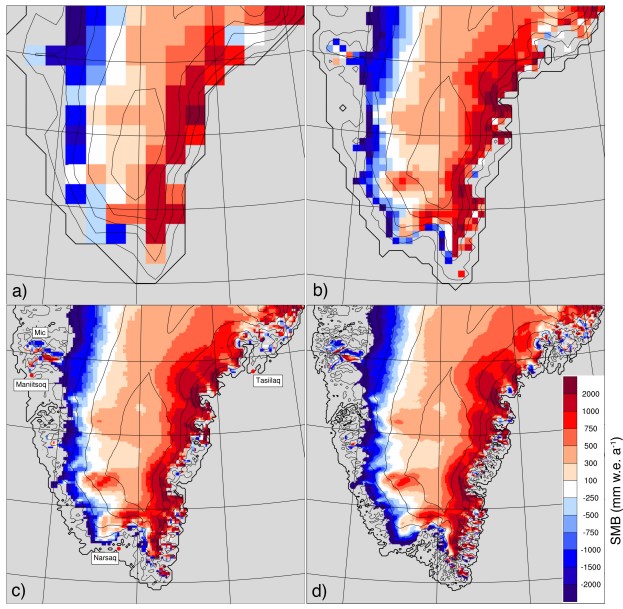

**Figure 2.** Annual mean SMB (mm w.e. $a^{-1}$) for the period October 2007 till September 2014 derived with RACMO2 with a resolution of **(a)** 60, **(b)** 20, **(c)** 6.6 and **(d)** 2.2 km, respectively. Grid points outside the ice sheet mask are grey. In **c)**, Mic is used as abbreviation for the Maniitsoq ice cap.

## 3 Results

### 3.1 Modelled SMB on various resolutions

Not surprisingly, the main characteristics of the GrIS SMB (Figure 2) show up in all resolutions, i.e. high accumulation along the eastern coast and an ablation zone along the western margin. However, finer structures start to emerge with increasing resolution, for example, wavy patterns in the precipitation field across topographic promontories, the windward and leeward SMB patterns over Maniitsoq ice cap (Fig. 2(c)) and the narrow ablation zones in the southern tip and along the East coast of Greenland. The differences between the modelled SMB field derived on the 2.2 and 6.6 km resolved grid seem minor but are significant at some locations, for example, west of Tasiilaq and north of Narsaq (Figure 2(c)).

The spatially integrated SMB for this part of Greenland varies with model resolution (Table 2). The low and medium resolution runs have an integrated SMB of about 70 Gt $a^{-1}$, but the 2.2 km resolution run deviates with an integrated SMB of 47 Gt $a^{-1}$. Apart from runoff, which slightly increases with resolution, however, no large change in the mean value of the different SMB components is found. Precipitation is rather similar for all resolutions, and although higher extremes are modelled for fine model resolutions, the standard deviation remains similar for all resolutions except for the 60 km resolution run. Up to 6.6 km resolution, the mean precipitation increases due to grid refining because resolving the topography induces that precipitation is concentrated on the ice sheet, which is higher than the surrounding areas, instead of around it. The decrease

**Table 2.** Mean and spatial variability of the annual SMB and its components in the investigated region in South Greenland. All values are in mm w.e. a$^{-1}$. Additionally, spatially integrated mean SMB is given in Gt a$^{-1}$.

| Resolution | Area | SMB | | Prp | SU | RU | ER$_{ds}$ |
|---|---|---|---|---|---|---|---|
| | 10$^3$ km$^2$ | Gt a$^{-1}$ | | mean / std; mm w.e. a$^{-1}$ | | | |
| 60 km | 317.1 | 64 | 203 / 911 | 836 / 825 | 34 / 30 | 598 / 863 | 1.1 / 4 |
| 20 km | 316.3 | 75 | 237 / 956 | 848 / 877 | 47 / 40 | 562 / 889 | 1.2 / 8 |
| 6.6 km | 318.4 | 71 | 223 / 1016 | 884 / 904 | 55 / 49 | 584 / 962 | 1.4 / 16 |
| 2.2 km | 319.7 | 47 | 148 / 1056 | 836 / 872 | 54 / 53 | 633 / 1061 | 1.6 / 31 |

of mean precipitation for the 2.2 km resolution run will be discussed in detail in Section 3.4. Sublimation is more than one order of magnitude smaller than precipitation and runoff. Mass loss by sublimation increases with resolution; at low resolutions the high sublimation rates due to the katabatic outflow of cold and dry air are not well modelled (not shown). The export of snow from the ice sheet by snow drift divergence is negligible at all resolutions. However, regional transport of snow becomes increasingly important at finer resolutions as the standard deviation of local mass changes increases by a factor 2 for each threefold increase of model resolution.

## 3.2 Statistical downscaling versus dynamical downscaling

Before discussing the comparison of the modelled SMB with observations, we analyse to which extent the SMB and its contributing components can be sensibly refined. This analysis is done for all six possible simulation pairs; results are discussed in detail for the downscaling of 60 km data to 6.6 km only. We focus on precipitation and runoff, because these two processes largely determine the SMB.

### 3.2.1 Precipitation

Figure 3 shows the results of refining precipitation fields from 60 to 6.6 km resolution. At 60 km resolution (Fig. 3a), the main characteristics of the precipitation distribution over Greenland are resolved. The eastern coast is wetter than the western coast; the higher interior is relatively dry and a precipitation shadow is located north of Maniitsoq ice cap (Fig. 2(c)). As the model grids of 6.6 and 60 km align perfectly by design, the nearest-neighbour interpolated precipitation field equals Figure 3a. Bilinear interpolation of this field to 6.6 km resolution (Fig. 3b) gives a smoother representation of the latter but does not contain additional information. Although the precipitation might seem to depend on elevation, applying statistical downscaling gives very similar results (Fig. 3c), because the correlation between precipitation and elevation is weak where the variability in precipitation is highest. In general, statistical downscaling provides lower estimates for (relatively) high altitude locations and higher estimates for low altitude locations (Fig. 3f). Statistical downscaling also enhances the precipitation shadow north of Maniitsoq ice cap. As a side note, precipitation estimates over the ocean, well away from Greenland, and derived with

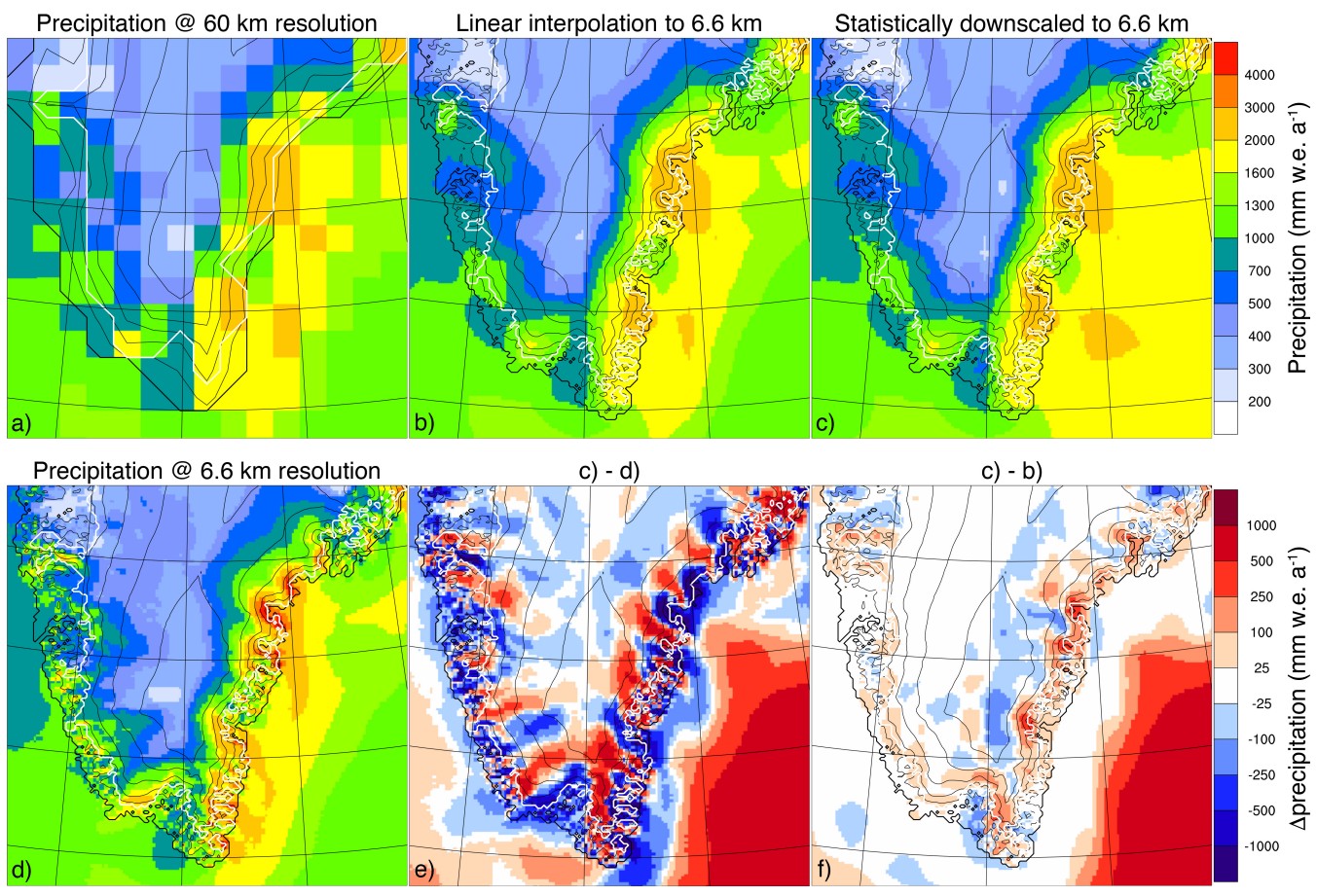

**Figure 3.** Results from refining precipitation from 60 km resolution to 6.6 km resolution. **a)** Modelled precipitation at 60 km resolution. **b)** Bilinear interpolated precipitation. **c)** Statistically downscaled precipitation using local regression to elevation. **d)** Modelled precipitation at 6.6 km resolution. **e)** Difference between statistically downscaled (c) and modelled (d) precipitation. **f)** Difference between statistically downscaled (c) and bilinear interpolated (b) precipitation. The white line mark the ice sheet margin.

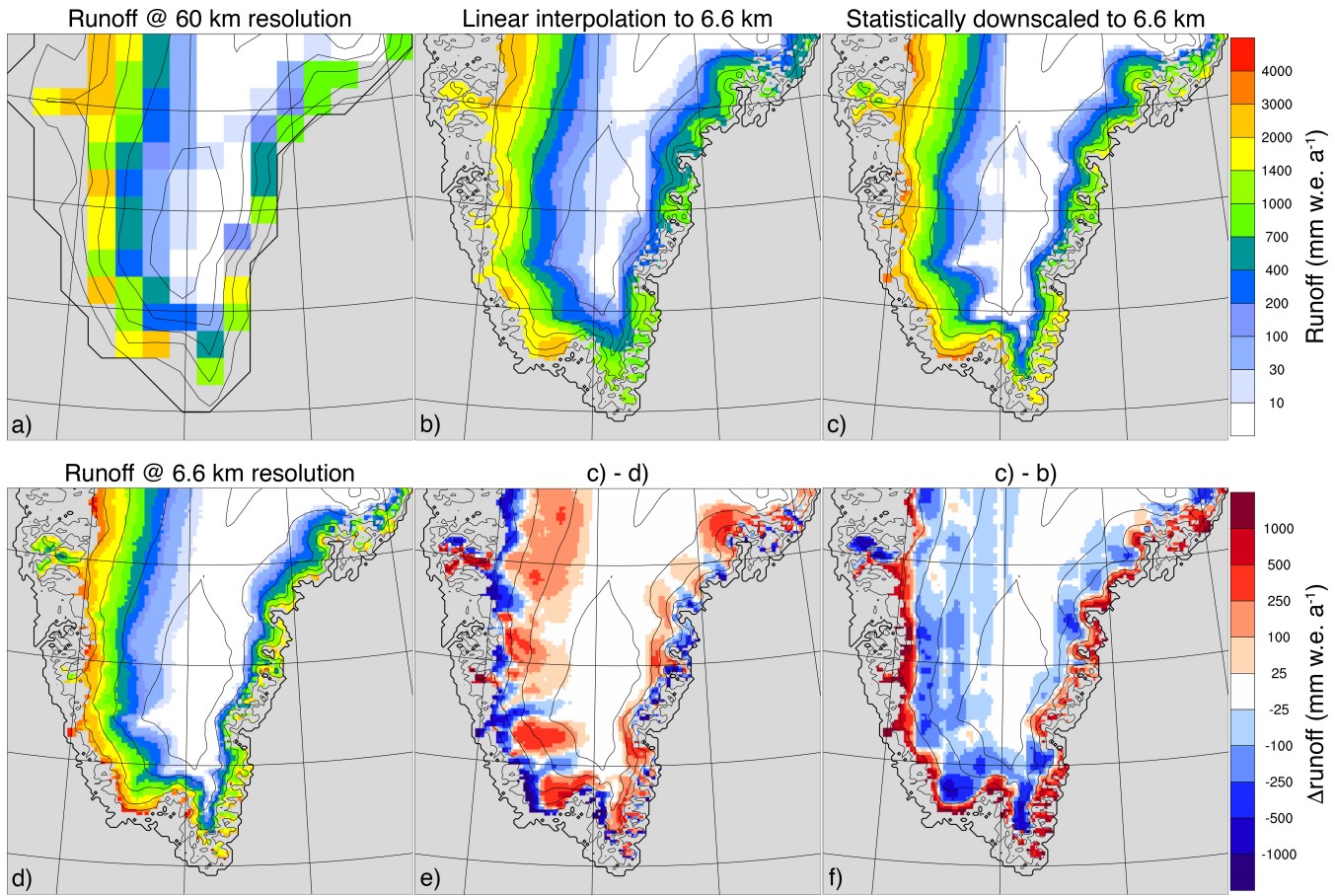

**Figure 4.** Results from refining runoff from 60 km resolution to 6.6 km resolution. **a)** Modelled runoff at 60 km resolution. **b)** Bilinear interpolated runoff. **c)** Statistically downscaled runoff using local regression to elevation. **d)** Modelled runoff at 6.6 km resolution. **e)** Difference between statistically downscaled (c) and modelled (d) runoff. **f)** Difference between statistically downscaled (c) and bilinear interpolated (b) runoff.

statistical downscaling deviate strongly as they are extrapolated from coastal precipitation rates since the statistical downscaling
procedures falls short away from Greenland in absence of topographic variations.

     However, all refining methods fail to reproduce the modelled precipitation patterns at 6.6 km resolution (Fig. 3d). At this resolution, along-coast variability is modelled: the windward side of promontories face high precipitation rates, while the leeward side of these promontories are much drier. These patterns are not reconstructed at all by statistical downscaling and the differences between estimated (Fig. 3c) and modelled (Fig. 3d) are very substantial (Fig. 3e).

### 3.2.2 Runoff

At 60 km resolution (Fig. 4a) a wide runoff zone is modelled for West Greenland and narrower zones for South and East Greenland. These zones are wider than modelled by Noël et al. (2015); 2006 - 2014 were relatively warm years. The additional runoff in the percolation zone is not buffered by refreezing since the firn layer had no time to build up in the simulations presented here. However, runoff estimates along the margins are robust since only winter snow is present every spring, hence the spin-up of the snow/ice pack is short. As for precipitation, a bilinear interpolation of the runoff fields to 6.6 km (Fig. 4b) is a smoothed version of the 60 km field. Note that no high runoff values are introduced along the margin. These higher runoff values are introduced if runoff is refined using statistical downscaling (Fig. 4c and f). At 60 km, maximum runoff estimates up to 2 m w.e. $a^{-1}$ are modelled, but with statistical downscaling the maximum runoff estimates increase to over 3 m w.e. $a^{-1}$. Higher runoff estimates are also introduced along the southern and eastern margins of the GrIS. Away from the ice sheet margin, statistical downscaling reduces the runoff due to the concave ice sheet topography. Due to the north-south orientation of the ice sheet, this introduces a striped difference pattern between linearly and statistically downscaled runoff (Fig. 4f). Numerical artefacts arise near the divide of the GrIS (Fig. 4c). Here, the correlation of runoff to elevation decreases, leading to negligible but incorrect patterns.

Comparing the downscaled patterns with the modelled runoff at 6.6 km resolution (Fig. 4d), statistical downscaling brings the estimate closer to the 6.6 km resolution runoff field compared to the bilinear interpolation but still lacks the high runoff values along the ice sheet margin. These high runoff rates occur because at lower elevation, ice is increasingly exposed during the summer, more summer precipitation falls as rain eliminating the mitigating effect of precipitation on melt and warm tundra air can flow to some extent onto the ice sheet. All these processes are not modelled at 60 km resolution, hence the refined product fails to reproduce the large runoff gradient near the ice sheet margin. Nevertheless, statistical downscaling provides much better runoff estimates than bilinear interpolation of fields. Note that this difference is not due to variations in the ice albedo, as a constant albedo is used. In reality the albedo varies greatly from place to place.

### 3.2.3 Statistical evaluation

Analysis of the refining techniques using Taylor diagrams (Fig. 5) and the RMSD (Table 3) provides a quantitative assessment of the expected ability of different refining techniques to reproduce spatial SMB patterns. The horizontal axis in Figure 5 displays the fraction of the true high-resolution model variability that is captured by the low-resolution model, while the vertical axis displays the amount of erroneous variability contained in the low-resolution model results. If a dataset matches perfectly with the reference dataset – excluding systematic biases – the normalised true and erroneous variability are one and zero, respectively. Therefore, this point on the $x$-axis has the label 'REF'. A dataset with similar variability as the reference dataset will be drawn on the dashed circle, if the variability is underestimated points will be displayed closer to the $(0,0)$ point. Figure 5a shows the reproduced variability of the SMB and its components for the three investigated techniques for refining from 60 to 6.6 km resolution. Crude nearest-neighbour interpolated fields are also assessed in this figure in order to quantify if bilinear interpolation is better than no efforts at all. For precipitation, bilinear interpolation removes spurious patterns since

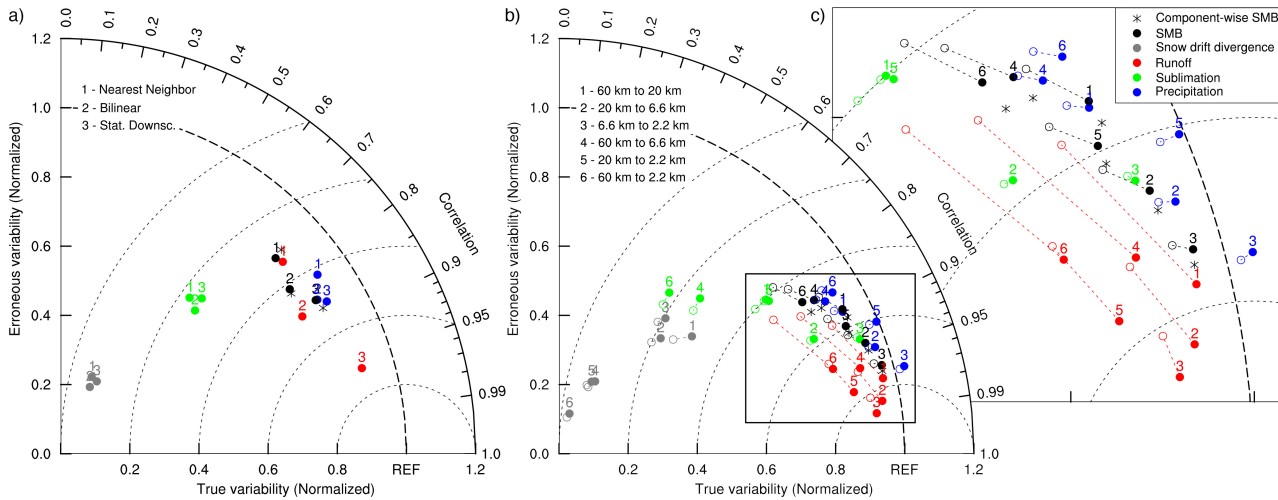

**Figure 5.** Taylor diagrams of the represented model variability for SMB and its components for **a)** various refining procedures from 60 to 6.6 km and **b)** bilinear (open circles) and statistical downscaling (filled circles) across all different model simulations, respectively, connected by dotted lines. **c)** Zoom of the boxed point cluster in b). The legend in c) applies to all panels. The horizontal location in the Taylor graph gives the fraction of the true variability reproduced by the model, the vertical location is the amount of erroneous variability as fraction of the standard deviation of the reference dataset.

the amount of incorrect information reduces, but bilinear interpolation does not add true model signal. Statistical downscaling adds some true model signal and provides, therefore, a better estimate of the modelled fields at 6.6 km, but this improvement in the representation is small. For runoff, statistical downscaling provides the best estimate even though bilinear interpolation also improves the representation of modelled runoff estimate. The statistical downscaling removes a significant amount of the incorrect deviations and adds a significant amount of true modelled variability. As a result, the refined runoff field correlates strongly with the modelled fields at 6.6 km resolution.

For the other SMB components, the results of the refinement are not as good as for runoff. The modelled sublimation patterns at 60 km resolution show far less absolute variability than the modelled sublimation at 6.6 km, and this variability is not added by statistical downscaling (Fig. 5a, green dots; Fig. S1). Statistical downscaling of the sublimation modelled at 60 km does also not reduce the general underestimation of sublimation compared to the 6.6 km run (Table 2). Sublimation fields obtained with bilinear interpolation or statistical downscaling are thus qualitatively not much better than sublimation fields produced by crude nearest neighbour remapping. This is even more the case for snow drift divergence (Fig. 5a, grey dots; Fig. S2). Patterns at 60 km resolution have significantly less variability than patterns at 6.6 km resolution and lack the ability to reproduce any of the patterns at 6.6 km resolution. Hence, refining makes little sense. However, since snow drift divergence is only a minor contribution to SMB, this has little impact on the ability to refine SMB.

Finally, the representation of high-resolution SMB estimates from low-resolution SMB estimates improves through application of statistical downscaling (Fig. 5a). Component-wise statistical downscaling of SMB (black star adjacent to black dot

**Table 3.** Root mean square differences (RMSD, mm w.e. a$^{-1}$) of bilinear (bl) and statistically downscaled (sd) variables for all resolution combinations. SMBc denotes component-wise processed SMB.

| Resolution | | SMB | SMBc | Prp | RU |
|---|---|---|---|---|---|
| Increase factor | | bl / sd | bl / sd | bl / sd | bl / sd |
| 60 km to 20 km | 3 | 487 / 429 | 472 / 404 | 306 / 296 | 345 / 185 |
| 20 km to 6.6 km | 3 | 383 / 342 | 367 / 318 | 224 / 219 | 244 / 147 |
| 6.6 km to 2.2 km | 3 | 297 / 286 | 288 / 270 | 163 / 169 | 190 / 144 |
| 60 km to 6.6 km | 9 | 587 / 518 | 573 / 488 | 358 / 343 | 443 / 250 |
| 20 km to 2.2 km | 9 | 481 / 436 | 461 / 413 | 254 / 256 | 342 / 234 |
| 60 km to 2.2 km | 27 | 650 / 562 | 636 / 519 | 350 / 336 | 539 / 324 |

#3) provides slightly better estimates of the SMB than statistical downscaling of the SMB itself (black dot #3). The differences are not large and enter through the constraints that are introduced in the refining step of individual components. For example, precipitation and runoff are both strictly positive, and precipitation is also modelled outside the ice sheet while runoff estimates are only derived on the ice sheet. The improvement of using component-wise statistical downscaling is likely larger if downscaling is applied on daily fields, when the local variations in precipitation and runoff are much more outspoken.

Figure 5b and Table 3 provides the results of refining all six possible resolution combinations. In Figure 5b, results of bilinear interpolation and statistical downscaling are shown with open and filled circles, respectively. In case of snow drift divergence (grey markers) statistical downscaling does not provide distinctly better results than bilinear interpolation. Moreover, the 6.6 km resolution data is as different to 2.2 km data as the 60 km resolution data is different to the 20 km resolution data, as points 1 to 3 cluster together. Hence there is no convergence in the spatial patterns modelled by RACMO2. Snow drift divergence is

largely driven by topography, with erosion at and upwind of crests, and deposition downwind of these crests (Fig. S2). Hence, a consistent relation with topography is absent and the magnitude of erosion and deposition patterns increases with finer resolution as the topographic features become rougher. Nonetheless, erosion and deposition patterns over the interior ice sheet are almost equal in the 6.6 and 2.2 km resolution runs. The poor performance of statistical downscaling is due to the very different patterns modelled along the mountainous ice sheet margins in the south-east (Fig. S2c, d and g) . Statistical downscaling also

hardly brings low-resolutions sublimation estimates (Fig. 5b, green markers) closer to high-resolution sublimation estimates compared to bilinear interpolation. However, the spatial pattern modelled at 6.6 km correlates well with the spatial patterns at 2.2 km (dot #3), so for sublimation a convergence to a definite spatial distribution starts to emerge, even though the 2.2 km has consistently less sublimation over the interior of the ice sheet and higher sublimation near the ice sheet margins (Fig. S1c, d and g). A similar convergence is found for precipitation and for downscaled and bilinearly interpolated runoff and SMB (Fig. 5b).

In all these cases, refined fields from 6.6 to 2.2 km (dots #3) are better than refined fields from 60 to 20 km (dots #1) with respect to resolved variability and remaining deviations (Tab. 3).

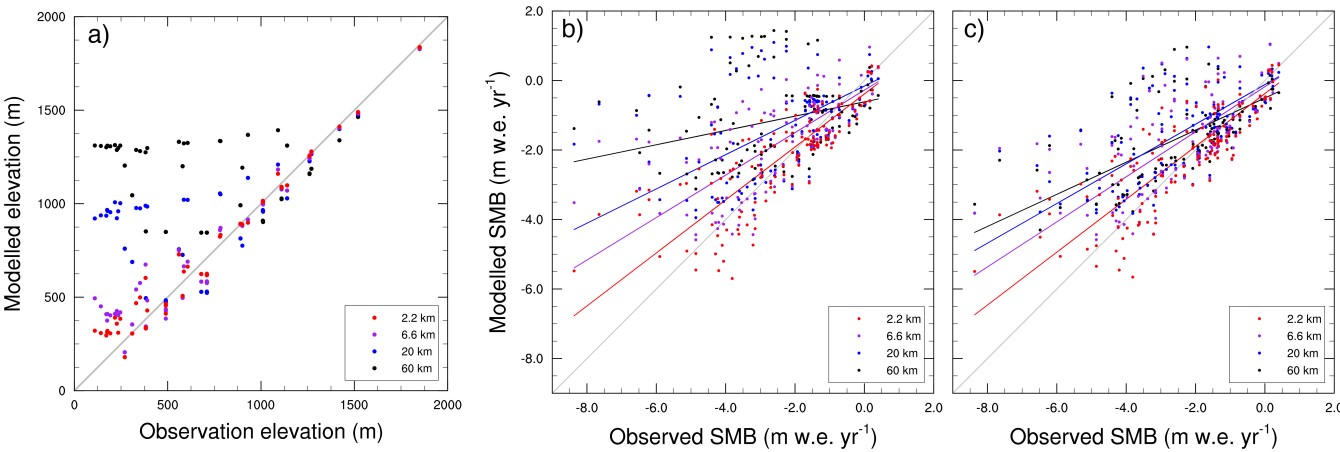

**Figure 6.** Estimated versus **a)** true observational site elevation and **b, c)** observed SMB for all four model resolutions using **a, b)** bilinear interpolation and **c)** component wise statistical downscaling, respectively. Fits are derived using orthogonal regressions and the 1-to-1 line is drawn in grey for reference.

### 3.3 Evaluation against observations

In the preceding section, it is shown how and when statistical downscaling is expected to improve model results. In this section, we assess if statistical downscaling indeed increases the agreement with observations. For this aim, model data is interpolated
or downscaled to the specific location of each observation.

First, model results are assessed using the "annual ablation" data, i.e. (sub)-annual, exactly time-matched ablation (Sec. 2.4). Table 4 shows that statistical downscaling, and especially component-wise statistical downscaling strongly improves low-resolution results. In more detail, the correlation of bilinearly interpolated SMB estimates from 60 and 20 km resolution runs is low and the regression slope is largely underestimated. As ablation is occurring along the GrIS edges, model data
is basically extrapolated to this edge to obtain a model estimate when refining low resolution model data. As a result, the apparent observation elevations - which are interpolated in an equal manner as SMB - are strongly overestimated (Fig. 6a), leading to a considerable underestimation of the ablation rates (Fig. 6b). This error, due to a too coarse model topography, is already largely removed in the 6.6 km resolution results, and application of statistical downscaling does not further improve the estimated SMBs. Noteworthy, the modelled SMB from the 2.2 km run clearly provides superior ablation estimates. This
will be analysed in more detail in section 3.4.

In Figures 7a and c, the results presented in Table 4 are graphically represented. Since a normal Taylor diagram (e.g. Fig. 7a) does not display systematic biases, Figure 7c has been added to include the contribution of the bias (horizontal axis) and the misrepresented variability (vertical axis) to the RMSD. In Figure 7a, the amount of misrepresented variability is the deviation of model results from the reference data (REF) and shown in the dotted circles around this 'REF'-point. Although RACMO2
overestimates the SMB (Fig. 6), the bias still is a smaller contribution to the RSMD than the misrepresented variability.

**Table 4.** Correlation, orthogonal regression slope, bias and RSMD for bilinear (bl), statistically downscaled (sd) and component-wise statistically downscaled (csd) SMB to the location of the observation, respectively.

| resolution | $r$ | | | regression slope | | | bias | | | RMSD | | |
|---|---|---|---|---|---|---|---|---|---|---|---|---|
| | bl / | sd / | csd | bl / | sd / | csd | bl / | sd / | csd | bl / | sd / | csd |
| | | | | | | | \multicolumn{3}{c}{m w.e. a$^{-1}$} | \multicolumn{3}{c}{m w.e. a$^{-1}$} | |
| 60 km | \multicolumn{3}{l}{0.19 / 0.43 / 0.57} | \multicolumn{3}{l}{0.21 / 0.50 / 0.46} | \multicolumn{3}{l}{1.31 / 1.01 / 0.80} | \multicolumn{3}{l}{2.30 / 1.94 / 1.65} |
| 20 km | \multicolumn{3}{l}{0.40 / 0.55 / 0.62} | \multicolumn{3}{l}{0.49 / 0.55 / 0.57} | \multicolumn{3}{l}{1.07 / 0.97 / 0.93} | \multicolumn{3}{l}{2.02 / 1.78 / 1.66} |
| 6.6 km | \multicolumn{3}{l}{0.63 / 0.62 / 0.62} | \multicolumn{3}{l}{0.61 / 0.66 / 0.65} | \multicolumn{3}{l}{0.67 / 0.67 / 0.68} | \multicolumn{3}{l}{1.53 / 1.55 / 1.56} |
| 2.2 km | \multicolumn{3}{l}{0.79 / 0.81 / 0.80} | \multicolumn{3}{l}{0.76 / 0.77 / 0.76} | \multicolumn{3}{l}{0.19 / 0.20 / 0.20} | \multicolumn{3}{l}{1.10 / 1.05 / 1.07} |

Furthermore, the reduction in the bias owing to the increase in model resolution or the application of statistical downscaling is larger than the decrease in the amount of missed variability (Fig. 7c). In both cases, high ablation locations are better resolved, which decreases the mean SMB and increases the variability. As the bias is relatively small for the best model results, the remaining source of error is the misrepresented variability in regional ablation. The contribution of misrepresented temporal variability is very small because the correlations and RMSDs of modelled SMB with the "period ablation" data, i.e. time aggregated "annual ablation" data, are very similar to the values shown in Table 4 for the "annual ablation" data (not shown).

When all available ablation observations are used, the resulting performance changes slightly (Circles in Fig. 7b and d). This difference arises from the improved spatial coverage of the ablation zones of South Greenland. Again, increasing the resolution as well as applying statistical downscaling both improve the results. Note that for 60 and 20 km model data component-wise statistical downscaling provides the best results, while for 6.6 and 2.2 km model data statistical downscaling of the modelled SMB gives slightly better results. Since component-wise statistical downscaling of SMB should theoretically outperform statistical downscaling of the SMB itself, this outcome may indicate that for the modelled SMB at 6.6 and 2.2 km resolution model shortcomings become more important than lack of resolution. Furthermore, although the 2.2 km simulation provides the results with the lowest RSMD, ablation is now overestimated as the bias has become negative (Fig. 7d). This overestimation could be due to the enhanced ablation in the most recent decades. The "average ablation" dataset includes many observations made between 1950 and 1980 when ablation was known to be less than in recent decades. In order to quantify the potential impact of this recent decrease in SMB on the bias, the "average ablation" dataset is also compared with modelled SMB of the RACMO2.3p2 simulation at 5.5 km (Noël et al., 2019) using both the full simulation length (1958-2018) and the specific time frame used here (2007-2014), showing that this temporal shift in SMB could be the same order of magnitude as the SMB biases of the statistically downscaled SMB estimates.

The behavior of model performance as function of resolution and refinement technique is very different in the accumulation zone (Fig. 7b and d, stars). 60 km resolution is insufficient to estimate accumulation as the variability is overestimated (1.25 to 1.4 times observed variability, Fig. 7b) due to erroneous variability that is of the same magnitude as the observed variability.

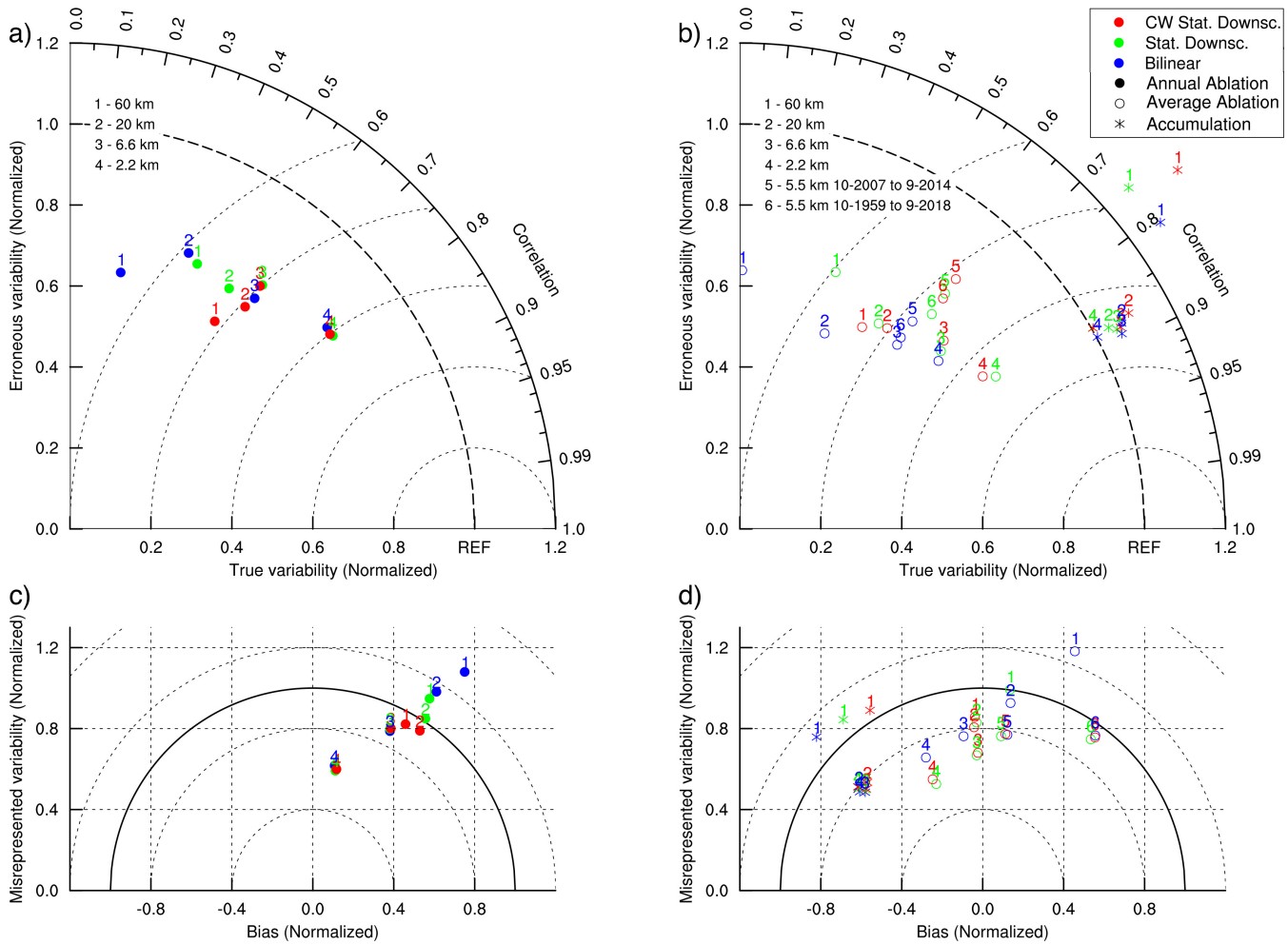

**Figure 7.** Graphical representation of the statistics of refined RACMO2 data for **a, c)** exact time-matching ablation observations, **b, d)** all ablation (circles) and accumulation (stars) observations, respectively. Subfigures **a, b)** are Taylor diagrams; subfigures **c, d)** show the contribution of the bias (horizontal axis) and misrepresented variability (vertical axis) to the RMSD (circles around $(0, 0)$). All data in this Figure are scaled with the standard deviation of the observation dataset. The 5.5 km data are taken from Noël et al. (2019). The legend in b), upper right corner, applies to all panels.

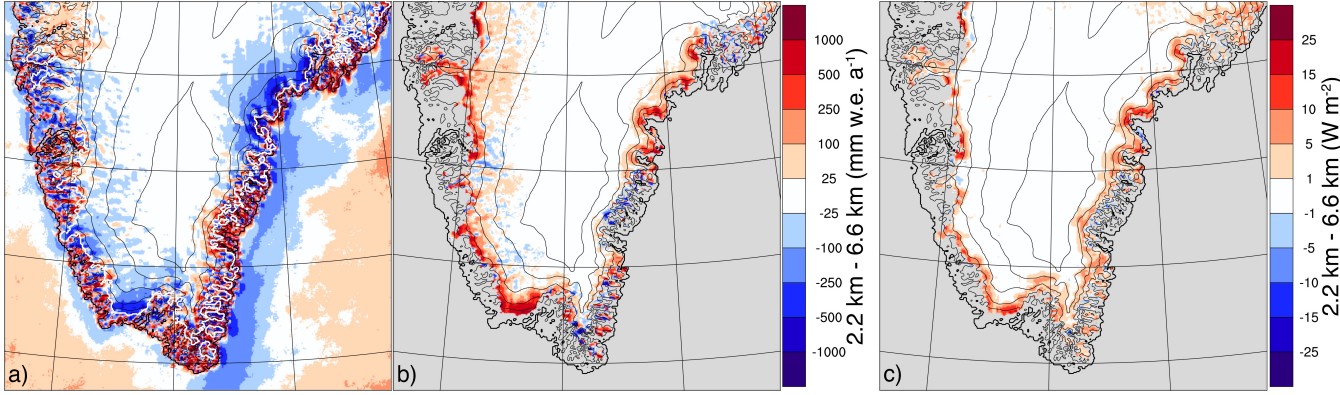

**Figure 8.** Differences in the mean **a)** precipitation, **b)** runoff and **c)** sensible heat flux between the 2.2 and 6.6 km simulations. The sensible heat flux is positive if pointed towards the surface. The 6.6 km results are mapped on the 2.2 km grid using bilinear (a) and statistical downscaling (b, c), respectively.

However, little to no gain in performance is seen between the 20, 6.6 and 2.2 km model results, and statistical downscaling has no added value, as the stars belonging to these resolutions are clustered in both panels. Of course, the possible impact of statistical downscaling was reduced by excluding observations with strongly deviating elevations. Furthermore, bilinear estimates are in most cases slightly better than statistically downscaled estimates. This result is in line with the earlier conclusion that statistical downscaling is not a suitable method for refining precipitation fields. We conclude that the quality of the modelled accumulation depends on the ability of the RCM or GCM to model precipitation patterns and that a resolution of about 20 km is enough to resolve the spatial patterns over the majority of the GrIS in South Greenland. Please note that in absence of observations, the modelled accumulation along the rugged southeastern margin of the GrIS cannot be evaluated.

### 3.4 Analysis of the 2.2 km run

There are three linked causes for the improved performance of the modelled SMB at 2.2 km resolution. They are discussed by comparing model results obtained at 2.2 km and 6.6 km resolution.

Firstly, less precipitation is modelled at 2.2 km than at 6.6 km resolution for most of the margin of the GrIS (Fig. 8a). Over the GrIS, these reductions are most outspoken West of Tasiilaq and North of Narsaq, where, by chance, also many of the ablation observations were located. In general, precipitation is reduced in the 2.2 km simulation compared to the 6.6 km simulation in coastal regions with relatively less rugged terrain. More precipitation is modelled over heavily rugged terrain and over sea at some distance from the coast. The differences are generally very small over the interior ice sheet, hence the performance representing the accumulation observations is similar.

The second and third cause are related to runoff as more runoff is modelled for the lower ablation zones (Fig. 8b). This is partly due to the lower precipitation as less precipitation reduces the melt water buffering capacity of the firn layer for refreezing in spring. Subsequently, less precipitation leads to earlier surfacing of dark glacial ice during the ablation season.

This results in similar patterns for precipitation and runoff. Runoff is further enhanced in the 2.2 km run by a higher sensible heat flux (SHF) towards the surface in the outer 20 km of the ice sheet (Fig. 8c). More detailed analysis (not shown) reveals that SHF is also increased during melt events, thus indeed enhances melt, and that both higher atmospheric temperatures and higher wind speeds contribute to the increase of SHF during melt events. Observations from automated weather stations (AWSs) point out that heat advection onto the ice sheet is a major driver of melt in the lower ablation zone, and the contribution of the SHF to melt is often underestimated in (regional) climate models (Fausto et al., 2016). However, it cannot be proven that the higher SHF is more realistic. An equivalent analysis of the 6.6 km results with respect to the 20 km results (Fig. S3c) reveals a similar increase of SHF, however spread over a threefold wider zone and a lower magnitude to at most 10 W m$^{-2}$. Since all model runs were performed with the same settings and similar time step lengths, the most probable reason for the increase of SHF is that a finer resolution allows for more dynamic heat advection onto the ice sheet at length scales up to ten times the grid resolution.

Such systematic differences in runoff and precipitation are absent when comparing the 6.6 km run with lower resolution runs (Fig. S3). Of course, topography induced periodic patterns arise in the 6.6 km run (e.g. Fig. 3d), but no regional decreases or increases of precipitation and runoff are found. These patterns were also not expected given the similar error margins compared to observations (e.g. Fig. 7a).

Figure S4 displays the differences between the modelled large-scale and convective precipitation at 2.2, 6.6 and 20 km resolution. Over the GrIS, more than 90% of the precipitation is large-scale precipitation, and the precipitation differences over the GrIS (Fig. 8a) are predominantly due to changes in modelled large-scale precipitation (Fig. 4c). A plausible explanation for the changes in large-scale precipitation over land is that the regional precipitation patterns are caused by the flow patterns which arise by operating a hydrostatic model over rugged terrain while using resolutions as fine as 2.2 km. In the 2.2 km run, standing 'supercritical flow'-like waves throughout the atmosphere are modelled fairly regularly over the mountainous coastal areas of South Greenland. These patterns are stronger over the more rugged coastal mountains (southeastern coast) and tundra zone (southwestern coast) than over the ice sheet as ice sheets strongly dampen topographic gradients due to the diffusive nature of ice flow. The momentum of the related vertical motion up to 10 m s$^{-1}$ is not conserved due to the applied hydrostatic assumption in RACMO2, which may lead to overestimated upward and downward motion. These waves induce (wet)-adiabatic cooling of air, subsequently extract humidity efficiently from the atmosphere through precipitation formation, which in turn leads to a reduction of available water vapour downstream and, therefore, to lower precipitation rates where topography favouring the generation of atmospheric waves is absent. These waves may or may not be realistic, but they will be better dealt with in a non-hydrostatic model.

Over sea, changes are due to different processes. Figure S4c shows that large-scale precipitation is rather uniformly reduced over the whole Atlantic sector in the model domain in the 2.2 km simulation. Similar reductions, however now for all ocean grid points, are visible between the 6.6 and 20 km simulation (Fig. S4e) and, to a lesser extent, between the 20 and 60 km simulation (not shown). No clear explanation nor reason has been found for this decrease. Away from the coast convective precipitation is enhanced in the 2.2 km simulation compared to the 6.6 km simulation (Fig. S4d). This enhanced convection is likely surface temperature driven, as this convective precipitation is not observed along the Greenland coast where sea surface

temperatures are lower due to the East Greenland Current and seasonal sea ice. Given the importance of convective precipitation over the Atlantic Ocean, the validity of the model results is seriously affected by the choice to keep the parameterization of convection equal for all model resolutions. Even though convection in these cold, shallow tropospheric atmospheric conditions may remain on kilometre scales, thus at a similar scale as the 2.2 km model grid resolution, it is disputable at least whether the used approach for convection is right.

Concluding, RACMO2 run at 2.2 km resolution provides superior results for the lower ablation zone due to changing precipitation patterns. Yet, these results may be in part be caused by the limitations set by the hydrostatic assumption in RACMO2 and questionable choices for convection. Nonetheless, It also shows that on coarser resolutions precipitation is a source of error. Finally, the absence of similar precipitation patterns in the 6.6 km run, nor in the RACMO2.3p2 5.5 km run compared to the RACMO2.3p2 11 km run (Noël et al., 2018; Noël et al., 2019, not shown) indicates that RACMO2 can be applied at resolutions down to about 5 km. Concluding, non-hydrostatic models can not be used in polar regions at resolutions below 5 km, even if convection is suitable represented.

## 3.5 Sensitivity of model performance on parameterisation choices

In the preceding sections the impact of resolution and of data refinement on the model performance has been discussed. For a perfect model, insufficient resolution would be the only source of errors. RACMO2 is, as any other model, not perfect, so for a certain resolution the intrinsic model errors would become of larger importance than the errors induced by the limited model resolution. In order to estimate intrinsic error due parameterisation and model initialisation choices in comparison to the changes differences induced by model resolution, eight additional full period runs on the 20 km domain have been carried out targeting four known possible sources of error: firn initialisation, snow densification, albedo and turbulent exchange of heat and moisture. Similar to the reference simulations, the analysis below, the first year of all simulations is excluded.

### 3.5.1 Firn initialisation

The reference runs at various resolutions were initialised with a spatially uniform fresh snow cover of 50 cm, leading to limited refreezing capacity in the accumulation zone. Figure 9a (run 1) shows the increase in SMB if was started with a spatially uniform layer of 10 m firn. A gradually increasing impact is visible from a small increase in SMB in the higher percolation zone up to 500 mm w.e. $a^{-1}$ higher SMB in the lower ablation zone. This spin-up effect lasts only for a few years, because if the first four years are discarded, hence excluding more vigorously firn spin-up effects, the SMB in the lower ablation zone (SMB $\ll$ -2 m w.e. $a^{-1}$) is comparable with the reference run (not shown). In the higher ablation and the percolation zones the SMB is still increased by up to 400 mm w.e. $a^{-1}$. The effect of initialising with an incorrect firn layer last longest close to the equilibrium line, which is also the region where the refreezing has also impact on the SMB.

These changes are not observed if the run is initialised with 3 m of firn (run 2, not shown); for this run the differences to the reference run are limited to about 20 mm w.e. $a^{-1}$. Hence, 3 m firn initialisation does not affect melt in the ablation zone for periods beyond two years, which is a positive effect, however, it also does not provide the required refreezing capacity in the

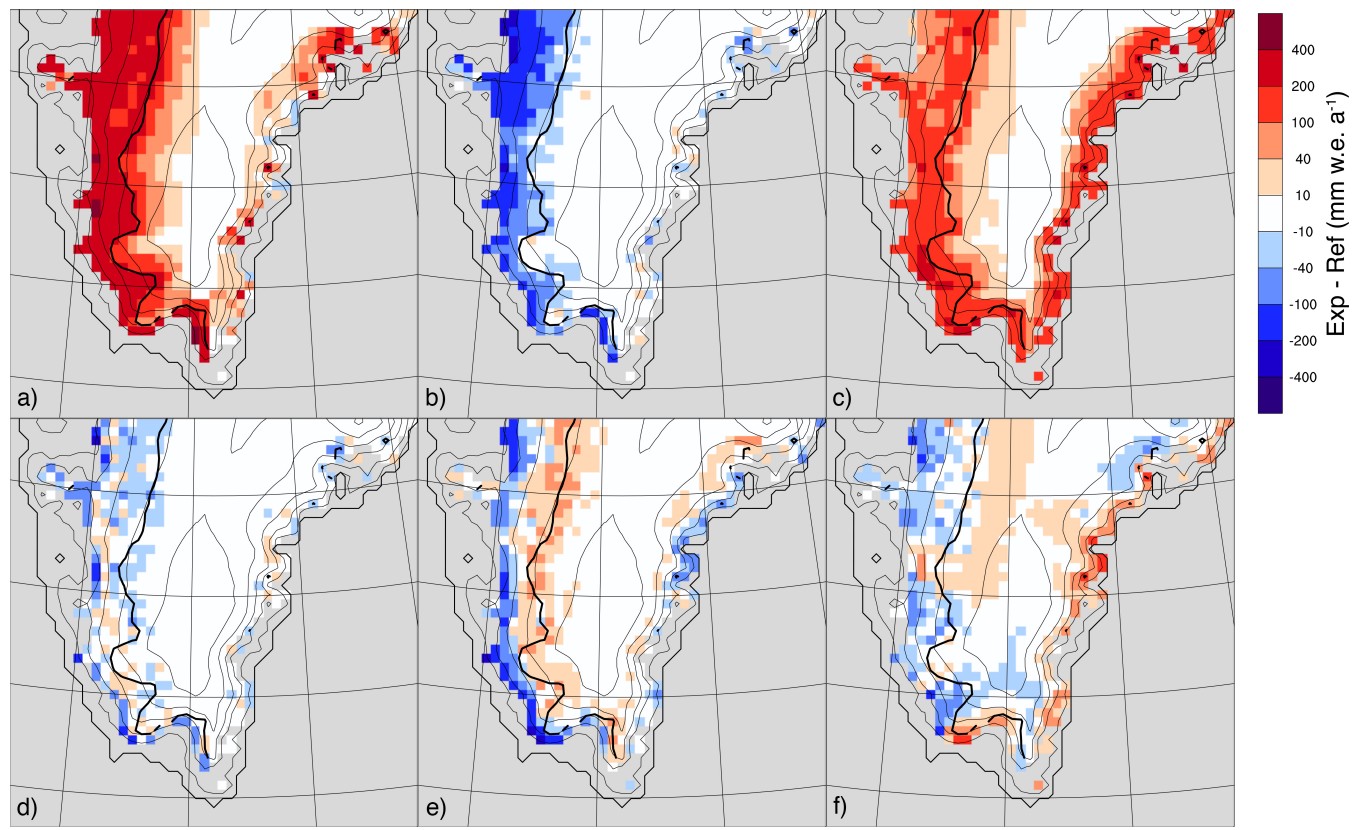

**Figure 9.** Difference in modelled SMB compared to the reference run from six of eight sensitivity test runs at 20 km resolution; October 2007 to September 2014 average: **a, run 1)** uniform 10 m firn layer, **b, run 4)** a reduced bare ice albedo by 0.04 to 0.38, **c, run 5)** the soot concentration in the snow halved to 0.05 ppmv, **d, run 6)** the roughness length of momentum of bare ice increased to 10 cm and **e, run 7; f, run 8)** equal roughness lengths of momentum, heat and moisture, with 1 cm over ice and (run 7) 1 mm or (run 8) 0.1 mm over snow, respectively. In all figures the equilibrium line in the reference run is drawn with a thick line for reference.

percolation zone, which is a negative effect. These runs show that in order to obtain accurate estimates of the SMB in both the percolation zone and the ablation zone, a proper initialisation of the firn/ice state is essential.

### 3.5.2 Firn compaction parameterisation

Next, a run is carried out using the densification parameterisation of Ligtenberg et al. (2011), fed with accumulation rates from
460 the reference run. Using Ligtenberg et al. (2011) yields an increase in melt of 10 to 50 mm w.e. $a^{-1}$ in the western ablation zone (run 3, not shown). This difference is due to the lower density of the winter snow, which reduces the ground heat flux in spring, allowing a slightly earlier onset of and subsequently stronger snow melt. Along the eastern coast, runoff is enhanced up to 150 mm w.e. $a^{-1}$. This enhancement is partly due to lower snow densities of near-surface snow and to faster compaction of

the high-accumulation firn layer, leading to a reduction in pore space for water retention and, therefore, to higher runoff rates
and lower SMB. Densification has thus a small but measurable impact on the SMB.

### 3.5.3  Albedo

Two runs were carried out to assess the impact of albedo on the SMB. In the first run (run 4), a uniform bare ice albedo of 0.38
is used, which is 0.04 lower than the value used in the reference simulations. A bare ice albedo of 0.38 is rather low but such
values are observed in several regions of the ablation zone, especially where dust and algae accumulate (Noël et al., 2018). As
expected, a lower ice albedo enhances melt and thus lowers the SMB up to 300 $\mathrm{mm}$ w.e. $\mathrm{a}^{-1}$ (Fig. 9b). Note that the lower
ablation zone, where the bare ice albedo influences SMB most, is underrepresented at this resolution, so the sensitivity of SMB
to ice albedo is probably higher at finer resolutions. This experiment reaffirms that it is important to have an accurate estimate
of the ice albedo.

In the second run (run 5), the soot content in snow is reduced from 0.10 to 0.05 $\mathrm{ppmv}$, which has a similar but opposite
effect on SMB (Fig. 9c) as the change in bare ice albedo. Less soot primarily increases snow albedo in winter and spring, which
delays the onset of melt and snow metamorphism, reducing the melt and runoff in the percolation zone. In the ablation zone,
lowering the soot concentration reduces runoff even more efficiently because when the onset of melt is delayed the moment of
bare ice surfacing is delayed as well. Hence, runoff is most sensitive to the soot content in snow in regions with both high melt
and high precipitation rates.

### 3.5.4  Turbulent exchange

Three runs were performed to examine the sensitivity of the SMB to the parameterisations of turbulent fluxes. Currently,
RACMO2 uses a roughness length for momentum ($z_{0m}$) of 1 $\mathrm{mm}$ over snow and 5 $\mathrm{mm}$ over bare ice and the roughness lengths
for heat ($z_{0h}$) and moisture ($z_{0q}$) are determined using Andreas (1987) over snow and Smeets and van den Broeke (2008b) over
bare ice. In run 6, $z_{0m}$ over bare ice is increased to 10 $\mathrm{cm}$, which represents a very rough terrain (Smeets and van den Broeke,
2008a). Although this is a 20-fold increase of $z_{0m}$, the effect on the modelled melt and thus on the SMB is limited (Fig. 9d).
In the parameterisations of Andreas (1987) and Smeets and van den Broeke (2008b), $z_{0h}$ and $z_{0q}$ are decreasing functions
of the roughness Reynolds number ($R_\star \equiv u_\star z_0/\nu$, with $u_\star$ and $\nu$ being the friction velocity and kinematic viscosity of air,
respectively). Increasing $z_{0m}$ increases the aerodynamic drag but also $R_\star$, which strongly reduces the increase of turbulent
heat and moisture exchange for higher $z_{0m}$.

It should be noted that Andreas (1987) uses the assumption of a neutral boundary layer for his derivation of $z_{0h}$ and $z_{0q}$
and the subsequent heat and moisture fluxes over snow. The variations of $z_{0h}$ and $z_{0q}$ are thus not only a surface property
but include flow properties as well, namely the limiting effect of the stable boundary layer on the turbulent fluxes. Smeets
and van den Broeke (2008b) correct for the limiting effect of the stable boundary layer on the turbulent exchange, but they use
relatively simple corrections (Pandolfo, 1966). Hence, $z_{0h}$ and $z_{0q}$ are still partly stable boundary flow properties and not solely
surface properties. In RACMO2, however, the limiting effect of the stable boundary layer on the turbulent fluxes is already
incorporated by using a modified version of the advanced gradient functions proposed by Holtslag and de Bruin (1988). Hence,

using both turbulence-dependent values for $z_{0h}$ and $z_{0q}$ and complex stability functions could lead to over-compensating the limiting effect of a stable boundary layer on the turbulent fluxes.

Therefore, two experiments were performed in which $z_{0h}$ and $z_{0q}$ were kept equal to $z_{0m}$. In run 7 (Fig. 9e) $z_{0m}$ was set to 1 and 10 mm over snow and ice, respectively. In this run the turbulent exchange is more efficient; the difference with the reference run is most prominent in the lower ablation zone where warm air reaches over the ice sheet. The enhancement of runoff in the ablation zone is larger than in the preceding sensitivity run (Fig. 9d) in which $z_{0m}$ over ice was increased to extreme values. Near the equilibrium line, the run shows that enhanced turbulent exchange generally results in reducing the melt and hence increasing the SMB. Finally, as we didn't aim to enhance exchange over snow, a run was executed in which $z_{0m}$ was set to 0.1 and 10 mm over snow and ice, respectively (run 8, Fig. 9f). In run 8, the turbulent exchange reduces slightly over snow, generally leading to more melt, compared to run 7, in the percolation zone. Along the margins of the ice sheet, snow melt is reduced and subsequently the SMB increases as the turbulent fluxes contribute less to removing the spring snow cover.

### 3.5.5 Sensitivity test runs evaluation

Figure 10 shows whether the set of test runs improve or deteriorate the comparison with observations. In many cases the differences with the reference run in predicting performance are very small, as the differences in modelled SMB are not large either. Including a firn layer, decreasing the soot content or fixing the roughness lengths (runs 1, 5 and 7) decrease runoff in the percolation zone, reducing the bias and amount of misrepresented variability in the accumulation zone (Fig. 1, red dots). However, the subsequent reduction of melt in the ablation zone in runs 1 and 5 decreases the performance in representing ablation (Fig. 1, magenta, blue and green dots). All other test runs have little impact on the accumulation zone and subsequently exhibit similar performance in representing the accumulation observations. Lowering the bare ice albedo (run 4) does change the bias in the ablation zone, however, the correlation of the model results with the observations does not increase. This test run shows, thus, that it is imperative to have an accurate spatial and temporal description of ice albedo as moderate changes (e.g. 0.04) have already a significant effect on the local SMB. The only change that generally improves the results is using constant $z_{0h}$ and $z_{0q}$ (run 7). This test run increases the gradient in the ablation zone, which is underestimated in all other test runs (Fig. 6c).

## 4 Discussion

This research, firstly, aims to find the optimal resolution at which global or regional atmospheric models must be run in order to produce realistic SMB estimates; secondly, to which extent statistical downscaling can replace full model simulations and, thirdly, which physical processes represent the largest sources of error in final SMB estimates.

Simulations performed at various resolutions indicate that high resolution dynamical downscaling, i.e. running an RCM, in combination with further refining through application of statistical downscaling, in principle provides the best estimate of the SMB in both the accumulation zone of southern GrIS and the ablation zone. Nevertheless, the results show that the performance gain by increasing the resolution flattens out. A resolution of 20 km is sufficient to resolve the accumulation zone of the GrIS;

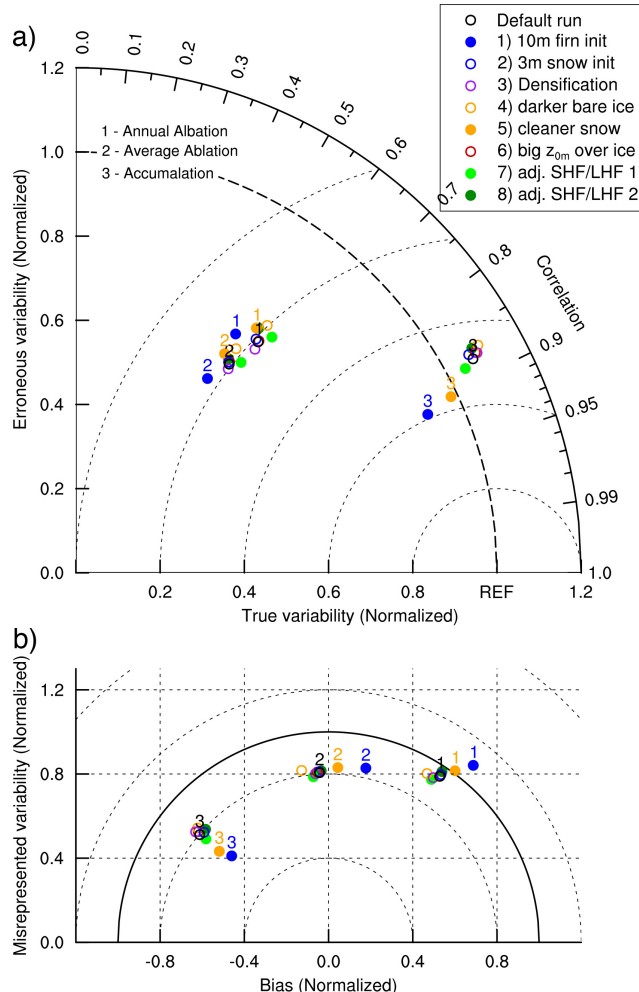

**Figure 10.** Sensitivity of the comparison of modelled SMB to observations as a function of snow initialisation, tuning and specific parameterisations, all derived using the 20 km resolution domain expressed by **a)** a Taylor diagram and **b)** RMSD contribution diagram like Fig. 7b and d.

6.6 km resolution is sufficient to resolve most of the ablation zone, and at 2.2 km almost all details of the GrIS and the larger peripheral glaciers are resolved. The regions of improvement represent a progressively smaller fraction of Greenland, however these regions near the margin exhibit, for example, the highest ablation rates. Nevertheless, the impact of the improvements induced by grid refinement on the overall SMB likely decreases as well (Lang et al., 2015; Fettweis et al., 2017). An exception could be the instance that the overall model behaviour changes due to increased model resolution. The runs at various resolutions and the accompanying methods of interpolation show that statistical downscaling only improves runoff estimates significantly; for all other SMB components the beneficial impact is marginal. As runoff is the dominating SMB component in the ablation zone, statistical downscaling does improve SMB estimates in such regions, especially if all components are refined separately. However, statistical downscaling cannot fully replace high-resolution dynamical downscaling simulations. Additionally, fractional glacier ice masks, which are not tested in this research, have shown to improve statistical downscaling as this approach provides also estimates of SMB components outside the glaciated domain.

The simulation at 2.2 km resolution also clearly show that the hydrostatic assumption in RACMO2 sets a lower mesh limit of about 5 km below which model results are no longer physically correct. Ironically, the regional shifts in precipitation patterns induced by the vertical air motion over topography lead to the best estimates of SMB. It also shows that changes in the precipitation flux in the ablation zone counts double as more snowfall increases the buffering capacity for refreezing and shortens the time that dark glacial ice is exposed at the surface.

Although we did not carry out further sensitivity tests with a direct focus on precipitation or clouds, processes involving these parameters can be significant sources of error in the modelled climate and subsequent SMB. Modelled precipitation patterns are harder to tune than ablation as local precipitation is the result of the interplay between the large-scale circulation and cloud microphysics. Noël et al. (2018) demonstrate that retuning the microphysics parameters in the cloud scheme of RACMO2 can change the ratio between the precipitation deposited inland and along the coast. Besides that, the microphysics currently used in RACMO2 is effectively not treating cloud water and cloud ice as separate prognostic variables; subsequently, for example, it underestimates the occurrence of cloud water over the ice sheet (Van Tricht et al., 2016). Furthermore, RACMO2 has no horizontal advection of hydrometeors, which leads to too pronounced precipitation maxima on the windward side of topography (Lenaerts et al., 2017; Agosta et al., 2019).

The sensitivity tests show that for optimal estimates a proper firn initialisation and spatially variable - and ideally also time varying - glacial ice albedos are essential; both are not used here. Remote sensing provides detailed estimates of the glacial ice albedo, however, there is not yet an established method to estimate the bare ice albedo and its evolution. Such a parameterisation will be very beneficial for simulations of past and future climate. Proper firn initialisations can be made using dedicated firn densification models; *a priori* knowledge of the location of the equilibrium line can be included if an initialisation for the present-day climate is prepared. However, even if those efforts are done, refreezing is most uncertain in the vicinity of the equilibrium line, where the refreezing capacity directly influences the SMB: the time in which a firn column adjusts is longest and the melt water percolation processes - ice lenses or aquifer formation - are very complex (Forster et al., 2013; Machguth et al., 2016a).

Compared to the impact of precipitation, firn initialisation and albedo, the impact of the parameterisation of the turbulent fluxes on the SMB is relatively small. Nonetheless, further research on the parameterisation of the turbulent fluxes is required, as our tests show that melt rates at the margins can be quite sensitive to parameterisation choices. Neither the default surface roughness parameterisation in RACMO nor the alternatively tested versions are to yield the best estimates of the turbulent fluxes. In order to make progress in the formulation of turbulent exchange under conditions of strong turbulent-driven melt dedicated flux measurements are required.

## 5   Conclusions

Concluding, the results show that with a dynamical model, i.e. an ESM, GCM or RCM, run at a resolution of 20 km ($\sim 0.25\,^\circ$) and subsequent statistical downscaling, reasonably good results for both the accumulation and ablation zone are obtained. An exception might be the rugged coastal accumulation zone of southeast Greenland where accumulation observations are missing. Yet, running a dynamical model at a higher resolution and subsequently applying statistical downscaling, leads to better results. In case of RACMO2, the best results are achieved with the 2.2 km simulation; a resolution, however, at which the hydrostatic assumption of RACMO2 is no longer valid. At this resolution local enhancement or reduction of precipitation formation due to topographically induced vertical motion is no longer negligible. Nonetheless, the modelled significant vertical motions at this resolution lead to better matching spatial precipitation patterns. Additional sensitivity tests show that using higher resolution is only one aspect in further improving the estimated SMB. Of approximately similar importance are a) a proper initialisation of the firn layer in the accumulation zone, b) the representation of the spatial and temporal variation of the albedo of glacial ice, c) correct estimates of the turbulent fluxes during melting events and d) the representation of the local and regional precipitation patterns. Only once all these aspects are properly resolved in a well balanced manner, the uncertainty and error in the modelled SMB of the GrIS can be further reduced.

*Data availability.*  Data sets presented in this study are available from the authors upon request and without conditions

*Author contributions.*  WJB prepared the manuscript, conducted the RACMO2 simulations and analysed the data. EVM and LHU provided assistance in setting up the simulations. All authors commented the manuscript.

*Competing interests.*  The authors declare no competing interests.

*Acknowledgements.*  Acknowledgement is made for the use of ECMWF's computing and archive facilities in this research. Horst Machguth is thanked for collecting and freely sharing his very relevant ablation observation dataset.

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
