# Peer review of "The added value of high resolution in estimating the surface mass balance in southern Greenland"

_The Cryosphere, 2019_

## Referee Comment (RC1) · Anonymous Referee #1 · 7 Jan 2020

Remarks to the Authors

Review of "The added value of high resolution in estimating the surface mass balance in southern Greenland" by Willem Jan van de Berg et al.

The Cryosphere Discuss. Manuscript Number: tc-2019-256

––––––––––––––––––––––––––––––––––––––––––––––––––––––––––––––––––––––––

General comments:

In this paper, the authors investigate and discuss dependency of the Greenland ice sheet (GrIS) surface mass balance (SMB) estimates by the polar regional climate

model RACMO2 on the choice of horizontal resolution set in the model (60, 20, 6.6, and 2.2 km). They highlight that setting a high horizontal resolution in the model is as important as making realistic initialization of firn physical conditions, simulating realistic ice albedo, estimating the accurate surface turbulent heat flux, and preventing positional displacement of precipitation systems in the model. This paper is very detailed, informative, and constructed well, so, this reviewer could enjoy reading it. However, I had a fundamental question about the authors' view on "the limitation of the hydrostatic assumption". I think the horizontal resolution of 6.6 km would be the maximum permissible limit for a hydrostatic atmospheric model like RACMO2, although I would set more than 10 km if I do the same things. In general, precipitating convective systems (its horizontal scale is not much wider than its vertical scale in general), which should be tried to resolve explicitly if an atmospheric model's horizontal resolution becomes less than about 10 km, cannot be simulated realistically by a hydrostatic atmospheric model due to "the limitation of the hydrostatic assumption". The horizontal resolution of 2.2 km for a hydrostatic atmospheric model is obviously out of the application range. Therefore, I thought presenting/discussing results from the 2.2 km run is not a good idea. An important but indistinct point of this study is whether the authors use a convective parameterization scheme in RACMO2 or not. Only if the authors use such a scheme and they have tuned it for high-resolution (less than 10 km) simulations around the GrIS well, the 2.2 km run would be worth trying. In the following part, this reviewer gives specific comments. Please note that page and line numbers are denoted by "P" and "L", respectively.

––––––––––––––––––––––––––––––––––––––––––––––––––––––––––––––––––––––––––––

Specific comments (major)

P. 3, L. 45 ∼ 48: In recent years, several attempts to develop non-hydrostatic regional models that can calculate temporal evolution of the GrIS SMB have been made (Mottram et al., 2017; Niwano et al., 2018). Please consider indicating this point here.

Sect. 2.1: Do the authors use a convective parameterization scheme in RACMO2? If the authors use such a scheme in the model, do they use the same scheme/setting for all the simulations introduced here? This is an important point of this study, so please explain it here.

Sections 2.1 and 2.2: The authors recognize the limitation of the hydrostatic assumption (P. 3, L. 44); however, they apply a hydrostatic model (RACMO2) at very high resolutions of 6.6 and 2.2 km. The authors' intention here is not clear.

P. 8, L. 188 ∼ 189: Ettema et al. (2009) showed that precipitation rates over the GrIS can be increased for higher-resolution simulations with hydrostatic atmospheric models. However, in this study, the precipitation rate from the 2.2 km RACMO2 run is smaller than that from the 6.6 km RACMO2 run (Table 2). What is the main reason of this result? Please discuss.

P. 17, L. 332 ∼ 334: The result described here is interesting: more precipitation occurs over not GrIS but tundra in the 2.2 km run than the 6.6 km run (I feel the result indicated here is plausible). Could the authors discuss more about possible reasons for this (what happens in the model?) here? It would be informative for readers.

P. 17, L. 335 ∼ 336: Why does the not-small difference in precipitation rates from the 2.2 and 6.6 km runs (25 ∼ 100 mm w.e. a-1) occur also over sea? Is it related to activities of cyclones and/or frontal systems in the area? Please discuss.

P. 19, L. 366 ∼ 368: The authors' logic here is comprehensible basically; however, how can we understand the argument that RACMO2 can be applied at resolutions down to about 5 km? In general, precipitating convective systems (its horizontal scale is not much wider than its vertical scale in general), which should be resolved realistically if an atmospheric model's horizontal resolution becomes less than about 10 km, cannot be simulated/reproduced realistically by a hydrostatic atmospheric model: I understand it is a specific example of the limitation of the hydrostatic assumption. Is it OK to understand precipitations over Greenland are caused mainly by non-convective systems? I

think maybe no. In this regard, I assume the choice/setting of a convective parameterization scheme in RACMO2 plays a key role of the result (please also see my major comment at Sect. 2.1 indicated above).

––––––––––––––––––––––––––––––––––––––––––––––––––––––––––––––––––––––

Specific comments (minor)

P. 3, L. 52 ∼ 65: This reviewer understands that the RACMO2 simulation at the horizontal resolution of 60 km is assumed to be an ESM-equivalent run in this study. However, is it a reasonable assumption? Because RACMO2 is forced by the ECMWF Operational Analyses (P. 5, L. 117), its performance might be better than ESM simulations in general. Usually, ESM simulations do not refer to such spatially and temporally detailed and realistic information.

P. 4, L. 76: In RACMO2, does meltwater runoff occur only at the bottom of snowpack? If an ice layer formation is simulated in the internal snowpack, what happens in the model?

P. 5, L. 102: Is there any references for the empirical formulas (also constants listed in Table 1: Ek, kc, and ks)? If no, please describe more in detail about the basic/background science of the equations.

P. 6, L. 119 ∼ 120: It is better to indicate reasons why the 2.2 km RACMO2 run can be conducted without an intermediate RACMO2 simulation (double-nesting).

P. 6, L. 127: Please explain why "0.42" is chosen and set for the ice albedo in the model.

P. 11, L. 229 ∼ 231: At this point, the authors have not compared model simulation results with measurement data. Therefore, I think they should not use the words like "improve" and "underestimate" here.

P. 14, L. 283: This is a good explanation of the analysis conducted in Sect. 3.2.3.
Related to the above comment (P. 11, L. 229 ∼ 231), it is better to indicate at the beginning of Sect. 3.2.3 that the discussion in the subsection 3.2.3 is an "expectation" of the effects of the refining techniques.

P. 21, L. 406: Maybe, the soot concentration value of 0.10 ppmv is set by default in RACMO2 right? If so, what is the basis of the value?

————————————————————————————————————————————

Technical corrections

P. 1, L. 8: It would a bit difficult to understand the exact meaning of "resembled". Can the authors replace "resembled" with another word?

————————————————————————————————————————————

References

Ettema, J., van den Broeke, M. R., van Meijgaard, E., van de Berg, W. J., Bamber, J. L., Box, J. E., and Bales, R. C.: Higher surface mass balance of the Greenland ice sheet revealed by high‐resolution climate modeling, Geophys. Res. Lett., 36, L12501, https://doi.org/10.1029/2009GL038110, 2009.

Mottram, R., Nielsen, K.P., Gleeson, E., Yang, X.: Modelling Glaciers in the HARMONIE-AROME NWP model, Adv. Sci. Res., 14, 323–334, https://doi.org/10.5194/asr-14-323-2017, 2017.

Niwano, M., Aoki, T., Hashimoto, A., Matoba, S., Yamaguchi, S., Tanikawa, T., Fujita, K., Tsushima, A., Iizuka, Y., Shimada, R., and Hori, M.: NHM–SMAP: spatially and temporally high-resolution nonhydrostatic atmospheric model coupled with detailed snow process model for Greenland Ice Sheet, The Cryosphere, 12, 635–655, https://doi.org/10.5194/tc-12-635-2018, 2018.

————————————————————————

---

## Referee Comment (RC2) · Anonymous Referee #2 · 14 Jan 2020

General comments In this manuscript the authors take a detailed look at the impact of varying the horizontal resolution of the RACMO2 model (from 60km to 2.2km) on the Surface Mass Balance (SMB) of the southern portion of the Greenland ice sheet. Furthermore, they incorporate the impacts of dynamical vs statistical downscaling on the SMB and its various components. Additionally, they investigate whether the resolution is the most important aspect in a modelling study by conducting sensitivity test by varying other factors such as bare ice albedo, firn initialisation and turbulent flux parameterisations. This paper is thorough and detailed, and it is clear that the authors have thought a lot about the contents of the study. However, as stated numerous times throughout the manuscript, RACMO2 is a hydrostatic model which should not be

run (or with caution/parameterisations) at resolutions higher than 5km, but the authors run the model at 2.2km resolution and results from this run are included throughout the study. It is a curious choice that the authors have chosen to run at such high resolutions regardless of the results being 'no longer physically correct', and the lower boundary on resolution in hydrostatic models is not a new finding. It is also a worry that future studies could use this paper as a basis for continuing to use high-resolution hydrostatic modelling despite their limitations, purely as they still provide a good comparison to observations. The authors should provide more information about the model set up, specifically for the convective parameterisations if they are to keep the 2.2km results. The specific comments are outlined below:

Major suggestions: Pg 3, Ln 60: Can you justify why you have used the 2.2km resolution run when hydrostatic models do not hold for such high resolutions? See general comments above.

Pg 3, Ln 46: Please provide examples of non-hydrostatic modelling of Greenland and the Arctic. Specifically, the Polar WRF model has been used quite frequently over Greenland for various climate studies (e.g Hines & Bromwich, 2008., Hines et al. 2011., Duviver & Cassano 2013., Turton et al in discussion with Earth System Science Data). Similarly, Niwano et al (2018) used a non-hydrostatic model coupled with a snow model, and Mottram et al (2017) used the non-hydrostatic HARMONIE model.

Section 2.2: What timestep was used in the model for each resolution? Later in the manuscript you mention 'similar' timesteps, but what are they? What is the height of your lowest model level for each resolution? Will these have an impact on your turbulent flux estimations?

Section 3.2.3 onwards: Taylor diagrams can be difficult to immediately understand, and there are lots of information on them, including shapes, colours and numbers. Whilst you give a good description on how to interpret them, a lot of back and forth between the figures, description and results is required to fully compare results and figures.

Help the readers by pointing out features of the diagram within the text. For example, Pg 17, ln 321 you say 'little to no gain in performance is seen between the results'. It would be useful to follow this with 'as seen in Figure 7b by the clustering of numbers 2,3,4 of the * shape'. Or similar. The authors should go back through the text to see where this information is required.

Section 3.4, Pg 17, Ln 332 onwards: Why is precipitation reduced in the 2.2km runs? Do you parameterise convection in RACMO2 for any of your runs? Or change any of the setup between runs? It would be useful to include a sensitivity test of convective parameters and how this affects precipitation as it is both an important and erroneous component of the SMB.

Pg 18, Ln 342 to 350: This section reads a little confusing. At the start you say that higher temperature and wind speeds contribute to the increased SHF. Towards the end of the paragraph you say that heat advection from the increased resolution is responsible. You should combine these lines of thought, as I assume they are all related to the resolution. Or was the initial sentence (increase T and WS) only related to melt events, and the latter to the full period? Do you have any AWS observations of SHF to compare with? As you could then provide a more definitive assessment than 'SHF fluxes near the margin are probably realistic'.

Pg 19, Ln 367-369: Why do you go down to 2.2km resolution then? See general comments.

Minor suggestions:

P1 Abstract, Ln 2: please add the time period that you cover, in brackets is fine.

P1 Abstract, Ln 13: '25km is sufficient'- where do you get this value from when you use 60km and 20km, and I struggle to find a reference to 25km in the manuscript.

Figure 1: The purple and red dots on Figure 1c and d are very similar looking. Could you try a different colour, or fill the dots on one? Also, later in the text you refer to the

purple ones as 'magenta' which is a better description than purple.

Pg 4, Ln 89: Unstable and convective conditions are regularly observed in summer. E.g Cohen et al 2007, Box et al 2001 (PhD thesis).

Pg 6: Ln 133: 'all methods are applied on time-averaged quantities and not on daily SMB fields'. What is the output frequency of your runs, and what time-average was used for output?

Figure 2: October 2008- September 2014 is correct? Or should it say October 2007? If you do not include 2007, why?

Figure 3: Please add a label/marker for Sukkertoppen as it is relevant for section 3.2.1. Just on one subfigure is fine.

Figure 4 caption: Write out the full caption, as there is a lot of information on this figure. The white line marking the ice sheet margin is missing.

Pg 11, Ln 228: Also highlight Figure 4f for numerical artefacts.

Pg 12, Ln 246/247: Refer to figure 5a.

Pg 13, Ln 257: 'significantly less variability'- have you tested the difference in variability with statistical tests?

Pg 13, Ln 270 onwards: can you include a spatial plot (like Fig 3 or 4) of the snow drift differences, as it would be easier to interpret line 270 to 280, especially the 'very different patterns' (Ln 275).

Figure 7 caption: 'all data in this Figure is' should say 'all data in this Figure are'. Add information about the dashed lines/REF in the caption.

Pg 17, Ln 320: in reference to figures, include that the * markers are what the reader should be looking at here. Similarly, what on the Taylor diagram is showing that the 60km resolution is 'insufficient'.

Pg 18, Ln 357: Remove 'These waves may or may not be realistic' here as you say it later.

Pg 19, Ln 362: remove 'as already mentioned' if you change the above sentence.

Pg 19, Ln 376: Why do you exclude the first year?

Pg 19, Ln 381: Why do you exclude the first 4 years?

Pg 21, Ln 406: Where do these soot content values come from? Literature or observations perhaps?

Section 3.5.5 is missing- or the subtitle is in the wrong place. It would make sense that this should go before Line 439, starting with 'Figure 10 shows...'

Pg 22, Ln 461: Whilst I agree that a smaller fraction of Greenland is being improved, aren't these some of the most important regions in terms of the SMB, such as the coast and low-elevation zones?

Pg 24, Ln 481: treating is spelt wrong.

References: Hines, K. M., & Bromwich, D. H. (2008). Development and Testing of Polar Weather Research and Forecasting (WRF) Model. Part I: Greenland Ice Sheet Meteorology*. MWR, 136(6), 1971–1989. https://doi.org/10.1175/2007MWR2112.1

Hines, K. M., Bromwich, D. H., Bai, L.-S., Barlage, M., Slater, A. G., Hines, K. M., ... Slater, A. G. (2011). Development and Testing of Polar WRF. Part III: Arctic Land*. Journal of Climate, 24(1), 26–48. https://doi.org/10.1175/2010JCLI3460.1

DuVivier, A. K., & Cassano, J. J. (2013). Evaluation of WRF Model Resolution on Simulated Mesoscale Winds and Surface Fluxes near Greenland. Monthly Weather Review, 141(3), 941–963. https://doi.org/10.1175/MWR-D-12-00091.1

Turton, J.V, Mölg, T. and Collier, E. (in discussion) High-resolution (1km) Polar WRF output for 79N Glacier and the Northeast of Greenland from 2014–2018, ESSD journal.

Doi: 10.5194/essd-2019-194

Niwano, et al (2018) NHM–SMAP: spatially and temporally high-resolution nonhydro-static atmospheric model coupled with detailed snow process model for Greenland Ice Sheet, The Cryosphere, DOI: 10.5194/tc-12-635-2018

Mottram et al (2017) Modelling Glaciers in the HARMONIE-AROME NWP model, Advances in science and research. DOI: 10.5194/asr-14-323-2017

Cohen et al (2007) Boundary-layer dynamics and its influence on atmospheric chemistry at Summit, Greenland. Atmospheric environment, https://doi.org/10.1016/j.atmosenv.2006.06.068

---

## Referee Comment (RC3) · Anonymous Referee #3 · 3 Feb 2020

The authors present a thorough and systematic comparison of the performance of the RACMO2 regional climate model run over a South Greenland domain for a series of consecutively increasing resolutions (60, 20, 6.6 and 2.2 km). This is a very valuable undertaking and I cannot recall having seen this done for this or other models in this part of the world. This makes it particularly useful as a reference for other modeling groups and further experimental design. The authors also include a second half of the manuscript that aims to give context to the resolution-dependent results by testing the sensitivity of melt and SMB to changes in various physical parameterizations.

The manuscript is generally well-written and well-structured and deals with an inter-

esting and important subject. It was a pleasure to read, although it did become a bit difficult and long to read at times (see detailed comments below). I have only two real concerns (see below) but I believe that the authors can address this with proper disclaimers. I therefore suggest to accept the manuscript with only minor revisions.

Major comments

Normally, I would consider the 6.6 km run to be within the grey-zone for hydrostatic physics and the 2.2 km run should be expected to be well within the range where it could break down. The authors do comment on this (eg. L ~365), but concerns about running hydrostatic at 2.2 km should have a more prominent place, including in the abstract, introduction and model-setup section.

The second part of the study where certain physical parameterizations are experimented with is the weakest part of the paper. They do not follow naturally from the former part of the paper and do perhaps even blur a bit the picture from the first, very stringent and systematic half of the paper. Also I am not convinced that the particular processes that are chosen for this set of sensitivity tests are adequately argued for. I gather that this part has been included to provide some sort of comparison of the magnitude of the resolution-change effects, but if that is the case then this comparison should be made explicitly.

Minor comments/typos

L11: "almost as well"

L36: "can the SMB be measured"

L50: "type of statistical"

L51: drop "do" before "correlate"

L140: "coarse"

L184: west of Tasiilaq?

Fig 3/4: Consider adding texts to the panels labeling them. That way the figure can almost be read without reading the caption.

Fig 5 etc: Taylor Diagrams are tricky to read for the untrained and the authors do a pretty good job of explaining them. But they could help the reader even more along the way. Also, there are details within the cluster of symbols situated in the lower right corner of Fig 5b. I suggest to include a blow-up of this part of the figure to allow the details to be visible.

L269: drop "of" before "6.6"

Table 4: The caption says "downscaled" several times, but it does not say which resolution is downscaled to.

Caption to Fig 7, second line: "ablation (dots)" should this be "(circles)"?

Fig 7: For some reason, it took me a while to realize that the legend in the top right corner of panel b actually applies to all panels. Can it be placed differently to make this more obvious?

L386-388: Very complicated sentence.

L429: This over-compensation would only occur over ice where the Smeets and van den Broeke-formulation is used, right? Please make this clear.

L438: Why is SMB reduced if melt is reduced?

L451: What happended to the content of section 3.5.5?

L458: as->and

L481: treating L501: I don't really understand the "thus" in this sentence. Do you rather mean "i.e."?

---

## Author Comment (AC1) · 3 Apr 2020

**Authors responses on the referee comments on the manuscript**
**"The added value of high resolution in estimating the surface mass balance in southern Greenland"**
**Willem Jan van de Berg, Erik van Meijgaard and Bert van Ulft.**

First of all, we would like to thank the referees for their constructive and detailed comments which helped to improve the manuscript. We have tried to incorporate all comments as good as possible.

The biggest concern raised is the 2.2 km simulation carried out with the hydrostatic model RACMO2. This concern is valid, applying RACMO2 at this resolution is violating incorporated assumptions on atmospheric hydrostatics and the representation of convection. Indeed, this manuscript should not give any room to be an excuse or motivation to use hydrostatic models on resolutions below 5 km for any operational simulations. Conversely, it should provide groups using a non-hydrostatic model, which are inherently more computationally expensive, further arguments for their approach.

Nonetheless, we believe that using RACMO2 at 2.2 km resolution in this academic study is justified as there is no other method than discussing flawed results to demonstrate that overstepping the resolution indeed lead to incorrect model behavior.

The comments by the reviewer are displayed in purple and citations of the revised manuscript text are displayed in Italics. Page numbers and line numbers always refer to those in the unrevised discussion paper.

In this document first the comments of the reviewers are discussed in detail. Next, a comparison of model results with observations is given. This discussion is added here as it is requested by a reviewer, but deemed unsuitable to be used in the manuscript. Finally, the revised manuscript with track changes is added. New text is listed in blue, removed text in red.

**Reviewer 1:**

R: In this paper, the authors investigate and discuss dependency of the Greenland ice sheet (GrIS) surface mass balance (SMB) estimates by the polar regional climate model RACMO2 on the choice of horizontal resolution set in the model (60, 20, 6.6, and 2.2 km). They highlight that setting a high horizontal resolution in the model is as important as making realistic initialization of firn physical conditions, simulating realistic ice albedo, estimating the accurate surface turbulent heat flux, and preventing positional displacement of precipitation systems in the model. This paper is very detailed, informative, and constructed well, so, this reviewer could enjoy reading it.

A: Thanks for these positive words.

R: However, I had a fundamental question about the authors' view on "the limitation of the hydrostatic assumption". I think the horizontal resolution of 6.6 km would be the maximum permissible limit for a hydrostatic atmospheric model like RACMO2, although I would set more than 10 km if I do the same things. In general, precipitating convective systems (its horizontal scale is not much wider than its vertical scale in general), which should be tried to resolve explicitly if an atmospheric model's horizontal resolution becomes less than about 10 km, cannot be simulated realistically by a hydrostatic atmospheric model due to "the limitation of the hydrostatic assumption". The horizontal resolution of 2.2 km for a hydrostatic atmospheric model is obviously out of the application range.

Therefore, I thought presenting/discussing results from the 2.2 km run is not a good idea. An important but indistinct point of this study is whether the authors use a convective parameterization scheme in RACMO2 or not. Only if the authors use such a scheme and they have tuned it for high-resolution (less than 10 km) simulations around the GrIS well, the 2.2 km run would be worth trying. In the following part, this reviewer gives specific comments. Please note that page and line numbers are denoted by "P" and "L", respectively.

A: We agree with the reviewer that using a hydrostatic model for simulations at resolutions below 5 km is not a good practice. However, this paper is meant to reaffirm this limit by demonstrating that the model results include fundamental flaws. This message is in the main text somewhat clouded by the fact that the modelled SMB at 2.2 km is better than the SMB modelled at lower resolutions. Nonetheless, we believe that is worth to demonstrate that also for the polar conditions of the Greenland Ice Sheet, where mesoscale convection is virtually non-existent, the hydrostatic assumption is still invalid.

Concerning convection, parameterizations have not been altered nor switched off or on. We have not considered to change anything on the parameterization of convection for two reasons. Firstly, convective precipitation is relatively unimportant over the ice sheet, see Figure R1 in this document. Secondly, even if we had adopted the best approach in the representation of convection, the results would show to be physically invalid. Hence, any effort to adopt the parameterization of convection to represent only smaller scale motion would be in vain.

Nonetheless, to make both points more clear in the manuscript, it is now mentioned numerous times that the 2.2 km simulation is invalid. Furthermore, it is discussed in detail that the representation of convection leads to undesired model behavior. See the responses on the specific comments for these changes in the manuscript.

**Specific comments (major)**

R: P. 3, L. 45 _ 48: In recent years, several attempts to develop non-hydrostatic regional models that can calculate temporal evolution of the GrIS SMB have been made (Mottram et al., 2017; Niwano et al., 2018). Please consider indicating this point here.

A: Thanks for these suggestions. We added the following sentence at this point: *"Nonetheless, non-hydrostatic models like WRF, NHM-SMAP and HARMONIE are increasingly applied for glaciated regions, e.g. Hines and Bromwich (2008); Mottram et al. (2017); Niwano et al. (2018); DuVivier and Cassano (2013), but due to their non-hydrostatic dynamical cores their computational costs are higher than hydrostatic models."*

R: Sect. 2.1: Do the authors use a convective parameterization scheme in RACMO2? If the authors use such a scheme in the model, do they use the same scheme/setting for all the simulations introduced here? This is an important point of this study, so please explain it here.

A: We have used the default convective parameterization of IFS version cy33r1 with equal settings for all simulations. We agree with the reviewer that if we aimed to present as realistic as possible results at 2.2 km resolution, it would be major flaw. Obtaining as realistic results as possible is, however, not our intention. Our aim is to show that, even though the results may look good, the results are non-valid as the modelled atmospheric circulation violates the hydrostatic flow assumption. We do not analysed the existence of undesired interaction between the implicit and explicit representation of convection at this resolution as it was not deemed very relevant within this context. To clarify the choices around convection, we added to section 2.1: "*The parameterization of convection in the IFS physics uses an adapted version of the model presented by Tiedtke (1989). This module is used for all simulations, also for the run at 2.2 km resolution. The choice has drawbacks, as the parametization of convective clouds can start competing with the explicitly resolved mesoscale convective systems, reducing the quality of the model results. However, it should be kept in mind that this 2.2 km is run and discussed to show that RACMO2 is unsuitable for this resolution, irrespecitve of the quality of the modelled SMB. Furthermore, convection precipitation is of limited importance for the SMB the GrIS, where convection is generally weak and limited to summertime.*"

R: Sections 2.1 and 2.2: The authors recognize the limitation of the hydrostatic assumption (P. 3, L. 44); however, they apply a hydrostatic model (RACMO2) at very high resolutions of 6.6 and 2.2 km. The authors' intention here is not clear.

A: To express more clearly our intention of the various simulations, we added to the introduction the following sentences: "*The simulation at 2.2 km has been performed to investigate to which extent violating the assumptions made in RACMO2, thus hydrostatic atmospheric flow and that convective motion must be fully parameterized, deteriorate the model results. The results presented here, therefore, provide by no means any justification to apply a hydrostatic model on resolutions below 5 km on operational basis.*"

P. 8, L. 188 - 189: Ettema et al. (2009) showed that precipitation rates over the GrIS can be increased for higher-resolution simulations with hydrostatic atmospheric models. However, in this study, the precipitation rate from the 2.2 km RACMO2 run is smaller than that from the 6.6 km RACMO2 run (Table 2). What is the main reason of this result? Please discuss.

A: In hind view, only in the simulations presented by Ettema et al. (2009), this effect occurred very profoundly. The cause is that by resolving the topography in more detail, precipitation is concentrated on the ice sheet instead of more evenly distribution on and off the ice sheet. For the 2.2 km simulation, different mechanisms are active, as will be discussed later in the manuscript. To clarify this, we added at this point: "*Up to 6.6 km resolution, the mean precipitation increases due to grid refining because resolving the topography induces that precipitation is concentrated on the ice sheet, which is higher than the surrounding areas, instead of around it. The decrease of mean precipitation for the 2.2 km resolution run will be discussed in detail in Section 3.4.*"

R: P. 17, L. 332 _ 334: The result described here is interesting: more precipitation occurs over not GrIS but tundra in the 2.2 km run than the 6.6 km run (I feel the result indicated here is plausible). Could the authors discuss more about possible reasons for this (what happens in the model?) here? It would be informative for readers.

A: We did not calculate whether the tundra receives more or less precipitation in the 2.2 km simulation compared to the 6.6 km simulation. For sure is that the variability is much stronger enhanced than over the Greenland Ice Sheet. It is all related with the small scale orographic variability, which is much lower over the ice sheet than over the surrounding mountains and tundra region. To make this more clear, we added at this point: "*These patterns are stronger over the more rugged coastal mountains (southeastern coast) and tundra zone (southwestern coast) than over the ice sheet as ice sheets strongly dampen topographic gradients due to the diffusive nature of ice flow.*"

R: P. 17, L. 335 _ 336: Why does the not-small difference in precipitation rates from the 2.2 and 6.6 km runs (25 _ 100 mm w.e. a-1) occur also over sea? Is it related to activities of cyclones and/or frontal systems in the area? Please discuss.

A: It is due to the different response of the convective scheme. We added a discussion on change in the precipitation patterns over sea: "*Over sea, changes are due to different processes. Figure S4c shows that large-scale precipitation is rather uniformly reduced over the whole Atlantic sector in the model domain in the 2.2 km simulation. Similar reductions, now for ocean grid points, are visible between the 6.6 and 20 km simulation (Fig. S4e) and, to a lesser extent, between the 20 and 60 km simulation (not shown). No clear explanation nor reason has been found for this decrease. Away from the coast convective precipitation is enhanced in the 2.2 km simulation compared to the 6.6 km simulation (Fig. S4d). This enhanced convection is likely surface temperature driven, as this convective precipitation is not observed along the Greenland coast where sea surface temperatures are lower due to the East Greenland Current and seasonal sea ice. Given the importance of convective precipitation 425 over the Atlantic Ocean, the validity of the model results is seriously affected by the choice to keep the parameterization of convection equal for all model resolutions. Even though convection in these cold, shallow tropospheric atmospheric conditions may remain on kilometre scales, thus at a similar scale as the 2.2 km model grid resolution, it is disputable at least whether the used approach for convection is right.*"

R: P. 19, L. 366 _ 368: The authors' logic here is comprehensible basically; however, how can we understand the argument that RACMO2 can be applied at resolutions down to about 5 km? In general, precipitating convective systems (its horizontal scale is not much wider than its vertical scale in general), which should be resolved realistically if an atmospheric model's horizontal resolution becomes less than about 10 km, cannot be simulated/reproduced realistically by a hydrostatic atmospheric model: I understand it is a specific example of the limitation of the hydrostatic assumption. Is it OK to understand precipitations over Greenland are caused mainly by non-convective systems? I think maybe no. In this regard, I assume the choice/setting of a convective parameterization scheme in RACMO2 plays a key role of the result (please also see my major comment at Sect. 2.1 indicated above).

A: Convective precipitation is of minor importance over the Greenland Ice Sheet, the relative contribution ranges from 10% along to margins to 2%, see Figure R1 in this document. Over the ocean, convective precipitation is very substantial. In the simulation at 2.2 km resolution, the enhanced convective precipitation compared to the 6.6 km resolution simulation (Fig. S4d) demonstrates that the convective scheme indeed performs 'differently' at this low resolution. However, no similar difference is observed between the 6.6 and 20 km resolution simulation (Fig. S4f), which strengthens the conclusion that for this polar context, a resolution of 5 km is still appropriate. It should be noted that in these cold conditions, convective cells are smaller than in the mid-latitudes or near the equator.

[Figure]

**Figure R1:** Fraction of convective precipitation of the total precipitation for the **a)** 2.2 km and **b)** 6.6 km simulation.

**R1, Specific comments (minor)**

R: P. 3, L. 52-65: This reviewer understands that the RACMO2 simulation at the horizontal resolution of 60 km is assumed to be an ESM-equivalent run in this study. However, is it a reasonable assumption? Because RACMO2 is forced by the ECMWF Operational Analyses (P. 5, L. 117), its performance might be better than ESM simulations in general. Usually, ESM simulations do not refer to such spatially and temporally detailed and realistic information.

A: We can understand that this connection, thus that the 60 km simulation represents an ESM, can and will be made. However, this 60 km simply has arisen from to choice that the grid refinement step was a factor 3 and that the highest tried resolution was 0.02 degree. Furthermore, applying an even lower resolution would pose practical problems as RACMO2 cannot be run with very few model grid points. Nevertheless, a resolution of 60 km is indeed optimistic for ESM, for which now typically 1 degree resolution is used. However, it is reasonable to assume that within a decade this typical resolution has gone down to something like 0.25 degree resolution. Furthermore, several ESMs with adaptive grids exists, which allows to resolve the Arctic or Greenland with 10-20 km resolution without the numerical burden to apply this resolution for the whole Earth. Finally, some ESMs get impressively close to re-analysis driven model simulations, e.g. Noël et al, TCD, ([https://doi.org/10.5194/tc-2019-209](https://doi.org/10.5194/tc-2019-209))

R: P. 4, L. 76: In RACMO2, does meltwater runoff occur only at the bottom of snowpack? If an ice layer formation is simulated in the internal snowpack, what happens in the model?

A: RACMO2 employs the bucket method for meltwater flow in snow and firn. Hence, in absence of snow, all melt water runs off directly; in that case 'at the bottom of the snow pack' is somewhat inaccurate as there is no snowpack. As this specific comment, thus where the meltwater is removed from the firn column, is not very relevant, we removed it from the text. Furthermore, ice lenses are never impeding downward percolation. To clarify this, we added in 2.1: "*Melt water percolation is modelled using the bucket method; if ice lenses are modelled in the firn pack, these are treated as permeable.*"

P. 5, L. 102: Is there any references for the empirical formulas (also constants listed in Table 1: Ek, kc, and ks)? If no, please describe more in detail about the basic / background science of the equations.

A: Although there are no references for the specific equations used here, we not entirely conceived these equations out of nothing. They were adopted in an attempt to make the semi-physical equations presented by e.g. Arthern et al (2010), Appendix B, preform as well as the empirical equations of e.g. Ligtenberg et al. (2011). This rationale has been added to the manuscript: "*Therefore, we initially explored an expression for creep of consolidated ice with cylindrical pores (Arthern et al., 2010, Eq. (B1)),*

$$\frac{\partial \rho}{\partial t} = k_{c,A10}(\rho_i - \rho) \exp\left[-\frac{E_{c,A10}}{RT}\right] \sigma \frac{1}{r^2}.$$

*The variables used in this and following Eqs. are listed in Table 1. However, we were unable to tune this relation to match the modelled firn densities with snow density profiles from Antarctica (van den Broeke, 2008). We choose to focus on Antarctic firn cores for tuning as in the Antarctic interior melt, which significantly alter the properties of firn cores and the densification process, is not occurring. Specifically, this equation fails to represent both the fast densification of low density, fine grained snow under very weak overburden pressure and the slower densification once the snow is denser and coarser grained while the overburden pressure is orders of magnitudes*

*bigger. However, densification partly depends on the recrystallisation of snow, which leads to a net growth of the crystals. Therefore, we modified Eq. (2) to the following empirical formula "*

P. 6, L. 119 - 120: It is better to indicate reasons why the 2.2 km RACMO2 run can be conducted without an intermediate RACMO2 simulation (double-nesting).
A: We preferred single-nesting above double-nesting for a uniform model setup across the resolution analyzed. This has been added to the manuscript: "*Although this 11-fold grid refinement step is bigger than typically used for high-resolution studies, it was preferred here to a double-nesting approach, as the latter would inhibit comparing simulations covering near-similar domains.*"

P. 6, L. 127: Please explain why "0.42" is chosen and set for the ice albedo in the model.
A: 0.42 is a typical value for bare ice of the GrIS. This notion has been added to the manuscript: "*Finally, a constant ice albedo of 0.42 – a typical bare ice albedo value for the GrIS is used, instead of MODIS derived albedo as is used in Noël et al. (2015), in order to improve the comparability of the model results on various resolutions.*"

R: P. 11, L. 229 _ 231: At this point, the authors have not compared model simulation results with measurement data. Therefore, I think they should not use the words like "improve" and "underestimate" here.
A: We agree with the reviewer that we are discussing whether statistical downscaling can represent higher-resolution model simulations and not observations, which are estimates of the true SMB. So, indeed, in this section we do not show that refinement methods make the results "truer", but we do show whether downscaling techniques improve the representation of high-resolution model results by low-resolution results. We have changed the wording in 3.2. at several places to make this clear. For example, P13, L 261 is now: "*Finally, the representation of high-resolution SMB estimates from low-resolution SMB estimates improves through application of statistical downscaling.*" For all other wording changes we refer the reviewer to the track-changed manuscript attached below.

P. 14, L. 283: This is a good explanation of the analysis conducted in Sect. 3.2.3. Related to the above comment (P. 11, L. 229 _ 231), it is better to indicate at the beginning of Sect. 3.2.3 that the discussion in the subsection 3.2.3 is an "expectation" of the effects of the refining techniques.
A: that is correct. We changed the sentence into "… *provides a quantitative assessment of the expected ability* …"

R: P. 21, L. 406: Maybe, the soot concentration value of 0.10 ppmv is set by default in RACMO2 right? If so, what is the basis of the value?
A: 0.10 ppmv is indeed the default soot value in RACMO2. This value is chosen after tuning experiments with RACMO2.1 and copied as default in RACMO2.3p1, and hence used in this study too. To explain this, we added to Section 2.2 the following sentence: "*The reference soot concentration is set to 0.10 ppmv, equivalent to the default value in RACMO2.1 and RACMO2.3p1 (van Angelen et al., 2012; Noël et al., 2015).*"

Technical corrections
R: P. 1, L. 8: It would a bit difficult to understand the exact meaning of "resembled". Can the authors replace "resembled" with another word?
A: We changed it into "*resolved*".

**Reviewer 2:**
**General comments**
R: In this manuscript the authors take a detailed look at the impact of varying the horizontal resolution of the RACMO2 model (from 60km to 2.2km) on the Surface Mass Balance (SMB) of the southern portion of the Greenland ice sheet. Furthermore, they incorporate the impacts of dynamical vs statistical downscaling on the SMB and its various components. Additionally, they investigate whether the resolution is the most important aspect in a modelling study by conducting sensitivity test by varying other factors such as bare ice albedo, firn initialisation and turbulent flux parameterisations. This paper is thorough and detailed, and it is clear that the authors have thought a lot about the contents of the study. However, as stated numerous times throughout the manuscript, RACMO2 is a hydrostatic model which should not be run (or with caution/parameterisations) at resolutions higher than 5km, but the authors run the model at 2.2km resolution and results from this run are included throughout the study. It is a curious choice that the authors have chosen to run at such high resolutions

regardless of the results being 'no longer physically correct', and the lower boundary on resolution in hydrostatic models is not a new finding. It is also a worry that future studies could use this paper as a basis for continuing to use high-resolution hydrostatic modelling despite their limitations, purely as they still provide a good comparison to observations. The authors should provide more information about the model set up, specifically for the convective parameterisations if they are to keep the 2.2km results.
The specific comments are outlined below.

A: We would like to thank the reviewer for his constructive comments. We indeed would like to keep the 2.2 km resolution results, for the sole aim to show that such a resolution leads to violation of the hydrostatic assumption. Yes, there is a curious coincidence that the 2.2 km results perform best, but if the reviewer believes that the revised manuscript still leaves room to conclude by readers that using 2.2 km resolution with a hydrostatic model for over Greenland and it surroundings is a sound idea, we are happy to state the invalidity of the hydrostatic assumption even stronger. We refer the reviewer for specific changes to the discussion of specific comments and the attached manuscript with track changes.

**Major suggestions:**

R: Pg 3, Ln 60: Can you justify why you have used the 2.2km resolution run when hydrostatic models do not hold for such high resolutions? See general comments above.

A: The justification for trying is simply to inquire how wrong or right the model will be. The reason for discussing the results in this manuscript is that there are very few papers that this clearly demonstrate that violating the hydrostatic assumptions lead to unphysical model behavior. Yes, we use the model results more extensively than the very minimal; conversely, we are humbly aware that 'all models are wrong but some are useful' (George Box). To add justification and context, we added the following sentence at this point: "*The simulation at 2.2 km has been performed to investigate to which extent violating the assumptions made in RACMO2, thus hydrostatic atmospheric flow and that convective motion must be fully parameterized, deteriorate the model results. The results presented here, therefore, provide by no means any justification to apply a hydrostatic model on resolutions below 5 km on operational basis.*"

R: Pg 3, Ln 46: Please provide examples of non-hydrostatic modelling of Greenland and the Arctic. Specifically, the Polar WRF model has been used quite frequently over Greenland for various climate studies (e.g Hines & Bromwich, 2008., Hines et al. 2011., Duviver & Cassano 2013., Turton et al in discussion with Earth System Science Data). Similarly, Niwano et al (2018) used a non-hydrostatic model coupled with a snow model, and Mottram et al (2017) used the non-hydrostatic HARMONIE model.

A: Thanks for these additions. The following line has been added: *"Nonetheless, non-hydrostatic models like WRF, NHM-SMAP and HARMONIE are increasingly applied for glaciated regions, e.g. Hines and Bromwich (2008); Mottram et al. (2017); Niwano et al. (2018); DuVivier and Cassano (2013), but due to their non-hydrostatic dynamical cores their computational costs are higher than hydrostatic models."*

R: Section 2.2: What timestep was used in the model for each resolution? Later in the manuscript you mention 'similar' timesteps, but what are they? What is the height of your lowest model level for each resolution? Will these have an impact on your turbulent flux estimations?

A: We added information on the time step at this point in the manuscript: "*Runs at the four resolutions were carried out with a time step of 150, 150, 150/90 and 60 s, respectively. For the 6.6 km simulation, the smaller time step of 90 s was only used for months with high wind speeds which causes the Lagrangian advection scheme to fail at processor sub-domain boundaries. In the 2.2 simulation, this problem was mitigated by extending the shared-data rim around sub-domains from 6 to 8 grid boxes.*"
We also added information on the elevation of the lowermost model levels: "*The lowest model levels were at approximately 10, 30 and 90 m, respectively, above the surface.*" As this mesh has not been varied for the various resolution, there is also no potential impact on the resolved turbulence.

Section 3.2.3 onwards: Taylor diagrams can be difficult to immediately understand, and there are lots of information on them, including shapes, colours and numbers. Whilst you give a good description on how to interpret them, a lot of back and forth between the figures, description and results is required to fully compare results and figures. Help the readers by pointing out features of the diagram within the text. For example, Pg 17, Ln 321 you say 'little to no gain in performance is seen between the results'. It would be useful to follow this with 'as seen in Figure 7b by the clustering of numbers 2,3,4 of the * shape'. Or similar. The authors should go back through the text to see where this information is required.

A: We have gone through the text to improve this. In order to keep this document readable, we refer the reviewer to the track-changed manuscript added below. In the specific case mentioned by the reviewer, we have added: *"…. no added value, as the stars belonging to these resolutions are clustered in both panels*."

R: Section 3.4, Pg 17, Ln 332 onwards: Why is precipitation reduced in the 2.2km runs? Do you parameterise convection in RACMO2 for any of your runs? Or change any of the setup between runs? It would be useful to include a sensitivity test of convective parameters and how this affects precipitation as it is both an important and erroneous component of the SMB.

A: It is reduced as due to enhanced rain-out over sharp mountains and convection over warmer ocean, the air is drier, leaving less moisture available for precipitation elsewhere. As now discussed in the methods sections, the parameterization of convection is left equal. The simulation 2.2 km is meant for academic reasons; it shows than RACMO is not suitable for this resolution, and not only due to the parameterization of convection. A sensitivity test on 2.2 km with a correctly adjusted convective scheme will still not show that this resolution is sound as the topographic waves will remain. A sensitivity experiment in section 3.5 could be another option, but we did not chose for this option. Firstly, convective precipitation is of minor importance over the ice sheet (Fig. R1, Fig. S4). Secondly, retuning of all cloud-content-to-precipitation parameters has a much stronger impact than adjusting the convective precipitation only. Thirdly, we believe, but that is hard to prove without simulations, that the usage of a 3-species cloud scheme without explicitly separated cloud ice and cloud liquid water and that neglecting the horizontal transport of falling precipitation will change the modelled precipitation patterns more than that a single sensitivity test can reveal. Nonetheless, we agree with the reviewer that it should be stressed that many improvements can be gained by improving the precipitation.

R: Pg 18, Ln 342 to 350: This section reads a little confusing. At the start you say that higher temperature and wind speeds contribute to the increased SHF. Towards the end of the paragraph you say that heat advection from the increased resolution is responsible. You should combine these lines of thought, as I assume they are all related to the resolution. Or was the initial sentence (increase T and WS) only related to melt events, and the latter to the full period? Do you have any AWS observations of SHF to compare with? As you could then provide a more definitive assessment than 'SHF fluxes near the margin are probably realistic'.

A: As Figure 8c shows the difference in SHF for the whole period, but enhanced SHF is only directly affecting the SMB during melt, we needed to tests whether in the 2.2 km simulation, also higher SHF occurs during melt event, otherwise the patterns in Figure 8c have no effect on the SMB. This is indeed the case.

Concerning the wording, we tried to clarify the intention by rewording the specific sentences to: "*Runoff is further enhanced in the 2.2 km run by a higher sensible heat 390 flux (SHF) towards the surface in the outer 20 km of the ice sheet (Fig. 8c). More detailed analysis (not shown) reveals that SHF is also increased during melt events, thus indeed enhances melt, and that both higher atmospheric temperatures and higher wind speeds contribute to the increase of SHF during melt events. Observations from automated weather stations (AWSs) point out that heat advection onto the ice sheet is a major driver of melt in the lower ablation zone, and the contribution of the SHF to melt is often underestimated in (regional) climate models (Fausto et al., 2016). However, it cannot be proven that the higher SHF is more realistic. An equivalent analysis of the 6.6 km results with respect to the 20 km results (Fig. S3c) reveals a similar increase of SHF, however spread over a threefold wider zone and a lower magnitude to at most 10 W m$^{-2}$. Since all model runs were performed with the same settings and similar time step lengths, the most probable reason for the increase of SHF is that a finer resolution allows for more dynamic heat advection onto the ice sheet at length scales up to ten times the grid resolution.*"

Lastly, we dived into a comparison of model with AWS observations, for which we chosen to use PROMICE data from south Greenland ([www.promice.dk](www.promice.dk)). Although we conclude from this analysis that "*…, it cannot be proven that the higher SHF is more realistic*", we assessed that the analysis included too many unresolved uncertainties to be included in the manuscript or supplementary materials. We added the analysis, the methods and its discussion below.

R: Pg 19, Ln 367-369: Why do you go down to 2.2km resolution then? See general comments.

A: Shortly speaking, to demonstrate it indeed leads to non-physical model behavior. See general response.

**Minor suggestions:**

R: P1 Abstract, Ln 2: please add the time period that you cover, in brackets is fine.

A: Done

R: P1 Abstract, Ln 13: '25km is sufficient'- where do you get this value from when you use 60km and 20km, and I struggle to find a reference to 25km in the manuscript.
A: We cannot refute this, therefore, the value is changed to 20 km.

R: Figure 1: The purple and red dots on Figure 1c and d are very similar looking. Could you try a different colour, or fill the dots on one? Also, later in the text you refer to the purple ones as 'magenta' which is a better description than purple.
A: In Figure 1c there are only magenta dots, all red dots are in Figure 1d. The red dot-like structures along the margin in Figure 1c are disconnected ice-free or ice-covered regions. As all dots are have a white fill, we believe that these red structures, which are not white filled, are distinctly different to the magenta dots. We believe that the figure caption and Section 2.4 is sufficiently clear in which subfigure which dots are drawn, so we did not change the figure, figure caption and Section 2.4. However, if the reviewer continues to disagree on this point, we are surely willing replace the magenta by another point. We replaced "purple" by "magenta in the figure caption.

R: Pg 4, Ln 89: Unstable and convective conditions are regularly observed in summer. E.g Cohen et al 2007, Box et al 2001 (PhD thesis).
A: We agree on that point, but as these situations are much less common, we did not mention the used parametrization for unstable boundary layers. To avoid the impression that the model cannot handle these situations, we added the following sentence: "*In case of unstable conditions, the flux profiles of Dyer and Hicks (1970) are used.*"

R: Pg 6: Ln 133: 'all methods are applied on time-averaged quantities and not on daily SMB fields'. What is the output frequency of your runs, and what time-average was used for output?
A: Our apologies for the confusion. For all flux fields like SMB and its components, accumulated fields (at time-step level) are used, so the output frequency (mostly 6 hourly) is not very relevant. The time-average period is the period of consideration, e.g. October 2007 to September 2014. To avoid this confusion, the sentence has been changed to: "*All methods are applied on period-averaged accumulated quantities and not on daily accumulated SMB fields.*"

R: Figure 2: October 2008- September 2014 is correct? Or should it say October 2007? If you do not include 2007, why?
A: No, this was indeed wrong, it should say October 2007.

R: Figure 3: Please add a label/marker for Sukkertoppen as it is relevant for section 3.2.1. Just on one subfigure is fine.
R: We got beaten by the Danish/Greenlandic geographical name confusion. Sukkertoppen equals the Maniitsoq ice cap, already located in Figure 2c. However, as North of Maniisoq would give new confusions, an additional label has been added in Figure 2c.
R: Figure 4 caption: Write out the full caption, as there is a lot of information on this figure.
The white line marking the ice sheet margin is missing.
A: Done. In Figure 4 there is no white line as only data on the ice sheet is shown, so a white line is redundant.
R: Pg 11, Ln 228: Also highlight Figure 4f for numerical artefacts.
A: The striped patterns in Figure 4f are not artefacts. Away from the margins, the topography is already rather well represented at 60 km resolution, so details of the topography becomes important. As the ice sheet surface is convex, at the edges of the 60 km grid boxes, the ice sheet elevation is higher than the linear gradient. As the statistically downscaled runoff would be more-or-less equal to the linear interpolated runoff if the topography would change linearly, the convex shape of the ice sheet leads to lower runoff estimates. We added the following text: "*Due to the north-south orientation of the ice sheet, this introduces a striped difference pattern between linearly and statistically downscaled runoff (Fig. 4f).*"

R: Pg 12, Ln 246/247: Refer to figure 5a.
A: As a reference is already made to this figure in the preceding sentence, we added "in this figure" to the sentence: "*Crude nearest-neighbour interpolated fields are also assessed in this figure in order to quantify if bilinear interpolation is better than no efforts at all.*"

R: Pg 13, Ln 257: 'significantly less variability'- have you tested the difference in variability with statistical tests?

A: No, we did not test it, but it is simple to argue why it is definitely significant. The variance in the snow drift divergence doubles for each resolution increase step. The 60 km simulation has 88 ice points, so the 95% uncertainty range in the standard deviation is about 20% (see for example https://en.wikipedia.org/wiki/Standard_deviation). For all other simulations, for which the number of glaciated points is far higher, this uncertainty range is even smaller. So, it is very unlikely that for any of the resolutions, the snow drift variability is actually a factor 2 different than estimated, hence the difference is significant.

R: Pg 13, Ln 270 onwards: can you include a spatial plot (like Fig 3 or 4) of the snow drift differences, as it would be easier to interpret line 270 to 280, especially the 'very different patterns' (Ln 275).

A: Given the limited impact of snow drift divergence on the SMB, we believe this figure is not relevant enough for the main text. Therefore, we have created a supplementary figures document, in which we displayed this graph. And, for consistency, we provided similar graphs for sublimation too. References to these graphs have been added in the document where relevant. As the difference between linearly interpolated and statistically downscaled snow drift divergence is minimal, we have decided to show differences between linearly interpolated and higher-resolution results (Fig S2e-g) as these the subfigures then focus on the difference in modelled patterns, not on the inability of create these patterns by a downscaling technique.

R: Figure 7 caption: 'all data in this Figure is' should say 'all data in this Figure are'. Add information about the dashed lines/REF in the caption.

A: Thanks for spotting. As Figure 7 is the second plot with Taylor diagrams, adding this here would be a repeat of the caption of Figure 5. Nonetheless, we added at p12, L244: "*Therefore, this point on the x-axis is has the label 'REF'.*"

R: Pg 17, Ln 320: in reference to figures, include that the * markers are what the reader should be looking at here. Similarly, what on the Taylor diagram is showing that the 60km resolution is 'insufficient'.

A: In order to clarify this, the text is changed into: "*The behavior of model performance as function of resolution and refinement technique is very different in the accumulation zone (Fig. 7b and d, stars). 60 km resolution is insufficient to estimate accumulation as the variability are overestimated (1.25 to 1.4 times observed variability, Fig. 7b) due to erroneous variability that is of the same magnitude as the observed variability.*"

R: Pg 18, Ln 357: Remove 'These waves may or may not be realistic' here as you say it later. A: Done

R: Pg 19, Ln 362: remove 'as already mentioned' if you change the above sentence. A: Done

R: Pg 19, Ln 376: Why do you exclude the first year?

A: We did this also for the reference runs. This has not been stated very clearly in the manuscript, this has been added to section 2.2: "*Unless stated differently, the first year of the simulation is excluded to reduce the effect of firn layer spin-up on the modelled SMB.*" and the specific sentence in Sec. 3.5 is rewritten to "*Similar to the reference simulations, the analysis below, the first year of all simulations is excluded.*"

R: Pg 19, Ln 381: Why do you exclude the first 4 years?

A: To analyse the effect of firn layer model spin-up. To make this more clear, then sentence has been changed into: "*This spin-up effect lasts only for a few years, because if the first four years are discarded, hence excluding more vigorously firn spin-up effects, the SMB in the lower ablation zone (SMB $\ll$ -2 m w.e. a$^{-1}$) is comparable with the reference run (not shown).*"

R: Pg 21, Ln 406: Where do these soot content values come from? Literature or observations perhaps

A: It is derived by tuning in earlier studies with RACMO2, and this should have been stated in the description of the model. It is added now in Section 2.2. "*The reference soot concentration is set to 0.10 ppmv, equivalent to the default value in RACMO2.1 and RACMO2.3p1 (van Angelen et al., 2012; Noël et al., 2015).*"

R: Section 3.5.5 is missing- or the subtitle is in the wrong place. It would make sense that this should go before Line 439, starting with 'Figure 10 shows: : :'

A: True, and corrected. This misplacement of the section header had slipped in while optimizing the figure placing in the discussion paper.

R: Pg 22, Ln 461: Whilst I agree that a smaller fraction of Greenland is being improved, aren't these some of the most important regions in terms of the SMB, such as the coast and low-elevation zones?
A: That is surely true but in our view not fully balancing the reduction of the "area of change". The sentence has been adjusted to: "*The regions of improvement represent a progressively smaller fraction of Greenland, however these regions near the margin exhibit, for example, the highest ablation rates. Nevertheless, the impact of the improvements induced by grid refinement on the overall SMB likely decreases as well (Lang et al., 2015; Fettweis et al., 2017).*"

R: Pg 24, Ln 481: treating is spelt wrong. A: corrected.

**Reviewer 3:**
R: The authors present a thorough and systematic comparison of the performance of the RACMO2 regional climate model run over a South Greenland domain for a series of consecutively increasing resolutions (60, 20, 6.6 and 2.2 km). This is a very valuable undertaking and I cannot recall having seen this done for this or other models in this part of the world. This makes it particularly useful as a reference for other modeling groups and further experimental design. The authors also include a second half of the manuscript that aims to give context to the resolution-dependent results by testing the sensitivity of melt and SMB to changes in various physical parameterizations.
The manuscript is generally well-written and well-structured and deals with an interesting and important subject. It was a pleasure to read, although it did become a bit difficult and long to read at times (see detailed comments below). I have only two real concerns (see below) but I believe that the authors can address this with proper disclaimers. I therefore suggest to accept the manuscript with only minor revisions.
A: We thank the reviewer for the positive feedback. These proper disclaimers have been added, as discussed already extensively. Furthermore, by embedding the comments the manuscript became even longer, our apologies for that.

**Major comments**
R: Normally, I would consider the 6.6 km run to be within the grey-zone for hydrostatic physics and the 2.2 km run should be expected to be well within the range where it could break down. The authors do comment on this (eg. L _365), but concerns about running hydrostatic at 2.2 km should have a more prominent place, including in the abstract, introduction and model-setup section.
A: These disclaimers has been added at these points. As this rebuttal document is already length, we refer the reviewer to the attached track-changes manuscript to see the textual changes.
R: The second part of the study where certain physical parameterizations are experimented with is the weakest part of the paper. They do not follow naturally from the former part of the paper and do perhaps even blur a bit the picture from the first, very stringent and systematic half of the paper. Also, I am not convinced that the particular processes that are chosen for this set of sensitivity tests are adequately argued for. I gather that this part has been included to provide some sort of comparison of the magnitude of the resolution-change effects, but if that is the case then this comparison should be made explicitly.
A: We agree that the second part is less systematic than the first part. However, it is hard to make a complete and accurate list of possible model errors, and some tests, e.g. on changes in the precipitation scheme or turbulence scheme, could be cumbersome to implement. In order to make the intention of the second part clearer, we added on P19L373: "*In order to estimate intrinsic error due parameterisation and model initialisation choices in comparison to the changes differences induced by model resolution, eight additional full …*". We did not change the wording in the introduction (P3L61-64), as we deem these sentences are clear on the aim of these sensitivity experiments.

Minor comments/typos
R: L11: "almost as well". A: Adjusted
R: L36: "can the SMB be measured". A: Adjusted
L50: "type of statistical". A: Adjusted
L51: drop "do" before "correlate". A: Adjusted
L140: "coarse". A: Adjusted
L184: west of Tasiilaq? A: True, adjusted
R: Fig 3/4: Consider adding texts to the panels labeling them. That way the figure can almost be read without reading the caption.

A: Done as requested.

R: Fig 5 etc: Taylor Diagrams are tricky to read for the untrained and the authors do a pretty good job of explaining them. But they could help the reader even more along the way. Also, there are details within the cluster of symbols situated in the lower right corner of Fig 5b. I suggest to include a blow-up of this part of the figure to allow the details to be visible.

A: As Reviewer 2 requested something similar, we have made the references to graphs 5 and 7 more explicit. Furthermore, we added a blow-up of Figure 5b as Figure 5c. The result is not extremely graphically appealing, but it works good enough.

R: L269: drop "of" before "6.6". A: Done

R: Table 4: The caption says "downscaled" several times, but it does not say which resolution is downscaled to.

A: In Section 3.3 we use data downscaled to the locations of the observations, so the resolution of the downscaled product does not play a role as such a product is not used. To clarify this, we added in P14, L 284: *"For this aim, model data is interpolated or downscaled to the specific location of each observation."* And to the caption of Table 4: *"…SMB to the location of the observation, respectively."*

R: Caption to Fig 7, second line: "ablation (dots)" should this be "(circles)"? A: yes, adjusted.

R: Fig 7: For some reason, it took me a while to realize that the legend in the top right corner of panel b actually applies to all panels. Can it be placed differently to make this more obvious?

A: We kept the legend at the same place, but added to the caption (and the caption of Figure 5 in similar fashion): *"The legend in b) right corner applies to all panels."*

R: L386-388: Very complicated sentence.

A: The sentence is adjusted to: *"This is partly due to the lower precipitation as less precipitation reduces the melt water buffering capacity of the firn layer for refreezing in Spring. Subsequently, less precipitation leads to an earlier surfacing of dark glacial ice during the ablation season."*

R: L429: This over-compensation would only occur over ice where the Smeets and van den Broeke-formulation is used, right? Please make this clear.

A: No it is the other way around. In Andreas (1988), the flux-limiting character of stable boundaries is fully incorporated in decreasing values of $z_{0h}$ and $z_{0q}$, while in Smeets and van den Broeke (2008b) the stability functions already do limit the fluxes for equal values of $z_{0h}$ and $z_{0q}$ In RACMO2, however, fluxes are strongly reduced in stable conditions by the atmospheric stability profiles, so decreasing $z_{0h}$ and $z_{0q}$ simultaneously would lead to double counting of the effect of the stable stratification on the fluxes. To make this more clear, we added *"Hence, $z_{0h}$ and $z_{0q}$ are still partly stable boundary flow properties and not solely surface properties."*

R: L438: Why is SMB reduced if melt is reduced?

A: This is indeed err, it is corrected to: *"Along the margins of the ice sheet, snow melt is reduced and subsequently the SMB increases as the turbulent fluxes contribute less to removing the spring snow cover."*

R: L451: What happended to the content of section 3.5.5?

A: This section header ended up at the wrong place while typesetting the figures. It is corrected now.

R: L458: as->and. A: Adjusted
R: L481: treating. A: Corrected

R: L501: I don't really understand the "thus" in this sentence. Do you rather mean "i.e."?

A: Yes, adjusted.

**Comparison of modelled SHF with AWS observations**

[Figure]

**Figure R2:** *Comparison of (a-f) the estimated SHF from observations and the difference between the estimated and modelled SHF from the (g-l) 6.6 km and (m-r) 2.2 km resolution simulation as function of the temperature (horizontal axis, K) and the wind speed (vertical axis, m s⁻¹) at 6 Promice AWSs in South Greenland. As the wind observations are made at 3.1 m above the surface (in absence of snow cover), observed wind speeds are multiplied by a factor 1.25 to estimate 10 m wind speeds. Black contours in all panels show the point density (K⁻¹ (m s⁻¹)⁻¹), multiplied by 100, for (a-f) the AWS observation and (g-r) model estimates. Green contours in panels (g-r) show the point density for the estimated observed 10 m wind speed.*

**Observations**

Observed hourly temperature and wind speeds observations and estimated SHF are downloaded from http://promice.org/PromiceDataPortal/. The observed temperature and wind speeds are not corrected for deviations of sensor height from the reference heights of temperature (2 m) and wind (10 m) observations. As temperature observations are carried out, if snow cover is absent, at 2.6 m above the surface, these values are used as representative at 2 m. Wind observations are carried out 3.1 m above the surface, if snow is absent, therefore, these values are multiplied by a factor 1.25 to estimate 10 m wind speeds. This 1.25 is a modest estimate of the wind speed difference between these two levels for stable conditions. The estimated SHF values are taken from the dataset, these were derived using a surface roughness length of 0.001 m.

**Comparison approach**

Data between either 1 October 2007 or the start of the observation data series, and 1 October 2014 are analyzed. Excluded are instances that DMI already discarded observations and if the obseved wind speed is exactly 0 m s⁻¹. Data is subsampled to 3 hourly values, i.e. the data of 0, 3, … 21 UTC is used, as RACMO2 data of

instantaneous temperature, wind and SHF values is stored on three-hourly resolution. In this manner, only RACMO2 is used for those moments that valid observations are available, excluding any potential sampling bias. For each AWS station, the data of the most proximate grid point is used and of one upslope or downslope grid point. The data is linearly interpolate between these two points to estimate observations at the exact elevation of the AWS station, excluding elevation induced temperature biases.

The stations NUK_L, MIT and TAS_L were excluded as such a grid point pair of glaciated model points was not available on both model resolutions, others were excluded as the time series was too short.

**Results**

Figure R2 shows the estimated SHF as function of the air temperature and wind speed. For the AWS observations (panels a-f), the values are uncorrected for any possible variations in observation height; for the other panels (g-r) SHF are gathered as function of 2 m temperature and 10 m wind speed. The figure focuses for melting conditions, as this analysis is carried out to analyze differences in the modelled ablation.

As mentioned in the manuscript, in the 2.2 km simulation SHF fluxes are higher than in the 6.6 km simulation. This is reflected in the point density plots of KAN_L, NUK_U, QAS_L and QAS_U, for which the 2.2 km models slightly higher probabilities for conditions with strong winds and high temperatures. For KAN_M and KAN_U, which are more inland, the point densities are near equal, and for these two sites the modelled point densities are rather similar to the observed point density. For NUK_U, the 6.6 kilometer simulation is closer to observations, while for QAS_L and QAS_U the 2.2 km simulation is close to the observations. Striking is that both simulation model rather frequently an absence of the katabatic wind for QAS_L, thus 2 m temperatures over 277 K while wind speeds are below 3 m s$^{-1}$, which is in reality hardly observed. "Non-katabatic" melt events are modeled for and observed at QAS_U, albeit less often than modelled for QAS_L. Finally, the strength of the katabatic wind for melting conditions is overestimated by both resolutions for KAN_L.

The modelled SHF agree generally well with the estimated SHF fluxes, with exception for KAN_L, where fluxes are higher in RACMO2, and strong wind cases for QAS_L and QAS_U, for which RACMO2 estimates smaller SHF values than is estimated from AWS data. However, it should be noted that the estimated SHFs from AWS are no direct observations, hence the estimated values depend strongly on the choices made, for example, the used surface roughness length. As it not known if the values given lead to a closed surface energy balance, it is not clear whether these SHF estimates are correct, it is neither possible to conclude whether RACMO2 gives realistic estimates of the SHF or not. Direct observations of SHF are extremely sparse and, as far as we know, not available for the period for RACMO2 has been run for this study.

However, as the representation of the katabatic boundary layer for melting conditions in the 2.2 km simulation is not clearly better than in the 6.6 km simulation, we conclude that it cannot be proven that the SHF estimates of the 2.2 km simulation is likely better than the 6.6 km. This would have been plausible if the 2.2 km simulation has a clearly better representation of the boundary layer than the 6.6 km simulation.

---

## Referee Report (RR1)

Review of van de Berg et al 2020.
Dear editor and authors,

the authors have taken on board many of the suggestions from myself and the other two reviewers, and the manuscript has improved in terms of accuracy and explanation. I particularly appreciate the effort to highlight that the 2.2km resolution runs were experimental and to show that, despite producing a relatively accurate SMB, the results are not physically realistic. My concern that this paper could spark the use of high-resolution hydrostatic models has now been resolved with this additional effort. I also think the Taylor diagrams have become clearer to interpret through enlarging some areas (Fig 5) and by including more description in the text. There are now only one or two very minor corrections required. After this, I recommend publishing this manuscript. As a side note, I suggest that the authors are careful with their use of 'his/he' when talking about the reviewer (e.g 'We would like to thank the reviewer for his constructive comments'), as this review was anonymous, and I am actually female.

Pg3, Ln 49: remove 's' from models to read: 'This type of model'
Pg3, Ln 50: change 'yet' to 'currently'
Pg 8, Ln 193: reword second part of sentence to: 'with the abbreviated name in brackets'
Pg21, Ln 438, 'It' needs to be lowercase.
Pg21, Ln 442, suitable should be suitably.
Pg 22, ln 453, change to: shows the increase in SMB if *the run* was started with…

Check use of British or American spelling throughout. On page 19, line 371 'behavior' is spelt in the American way. Whereas page 21, line 434 'kilometre' is the British spelling.

---

## Author Response (AR2)

Authors responses on the referee comments on the manuscript
"The added value of high resolution in estimating the surface mass balance in southern Greenland"
Willem Jan van de Berg, Erik van Meijgaard and Bert van Ulft.

We would like to thank the two reviewers for their review and their judgement that the paper is now almost ready for publication. We sincerely apology to reviewer 1 for addressing her with "he".

**Reviewer 1:**

R: The authors have taken on board many of the suggestions from myself and the other two reviewers, and the manuscript has improved in terms of accuracy and explanation. I particularly appreciate the effort to highlight that the 2.2km resolution runs were experimental and to show that, despite producing a relatively accurate SMB, the results are not physically realistic. My concern that this paper could spark the use of high-resolution hydrostatic models has now been resolved with this additional effort. I also think the Taylor diagrams have become clearer to interpret through enlarging some areas (Fig 5) and by including more description in the text. There are now only one or two very minor corrections required. After this, I recommend publishing this manuscript. As a side note, I suggest that the authors are careful with their use of 'his/he' when talking about the reviewer (e.g 'We would like to thank the reviewer for his constructive comments'), as this review was anonymous, and I am actually female.

A: We would like to thank the reviewer for her positive comments.

Pg3, Ln 49: remove 's' from models to read: 'This type of model'
Pg3, Ln 50: change 'yet' to 'currently'
Pg 8, Ln 193: reword second part of sentence to: 'with the abbreviated name in brackets'
Pg21, Ln 438, 'It' needs to be lowercase.
Pg21, Ln 442, suitable should be suitably.
Pg 22, ln 453, change to: shows the increase in SMB if the run was started with…

A: We adopted all this.

R: Check use of British or American spelling throughout. On page 19, line 371 'behavior' is spelt in the American way. Whereas page 21, line 434 'kilometre' is the British spelling.

A: We chose to use British English.

**Reviewer 2:**

R: Thanks to the authors' detailed responses, this reviewer could understand the authors' intention and the purpose of this study. Now, I agree with the authors' opinion that showing results from the 2.2 km RACMO2 run in this study is justified and am happy to recommend its publication in TC. During the discussion, I found the results presented in this paper imply that the horizontal resolution ranging between 6 and 3 km lies in a "gray zone" in RACMO2 applied in the GrIS. I would like to recommend the authors to add a statement on the "gray zone" of

RACMO2, which can be an additional conclusion of this study, in the final version of the manuscript.

A: Mentioning a grey zone between 6 and 3 km creates an ambiguity we would not like to make. We prefer to stick to the 5 km limit as mentioned now, as this is also our experience from other simulations. For example, we have carried out a 3.7 km simulation over Svalbard, but analysis showed that the results were unpublishable for similar reasons as discussed in this manuscript.

[revised manuscript text omitted]